# Commissureless acts as a substrate adapter in a conserved Nedd4 E3 ubiquitin ligase pathway to promote axon growth across the midline

Kelly G Sullivan, Greg J Bashaw*

Department of Neuroscience, Perelman School of Medicine, University of Pennsylvania, Philadelphia, United States

## eLife Assessment

This work is of **fundamental** significance to the field of nervous system development as it advances our mechanistic understanding of axon guidance. The rigorous biochemical and genetic approaches are **compelling**, experiments are well-controlled, and the major claims are supported by **convincing** data. The study should be of general interest to the developmental neurobiology community.

**\*For correspondence:**
gbashaw@pennmedicine.upenn.edu

**Abstract** In both vertebrates and invertebrates, commissural neurons prevent premature responsiveness to the midline repellant Slit by downregulating surface levels of its receptor Roundabout1 (Robo1). In *Drosophila*, Commissureless (Comm) plays a critical role in this process; however, there is conflicting data on the underlying molecular mechanism. Here, we demonstrate that the conserved PY motifs in the cytoplasmic domain of Comm are required allow the ubiquitination and lysosomal degradation of Robo1. Disruption of these motifs prevents Comm from localizing to Lamp1 positive late endosomes and to promote axon growth across the midline in vivo. In addition, we conclusively demonstrate a role for Nedd4 in midline crossing. Genetic analysis shows that *nedd4* mutations result in midline crossing defects in the *Drosophila* embryonic nerve cord, which can be rescued by introduction of exogenous Nedd4. Biochemical evidence shows that Nedd4 incorporates into a three-member complex with Comm and Robo1 in a PY motif-dependent manner. Finally, we present genetic evidence that Nedd4 acts with Comm in the embryonic nerve cord to downregulate Robo1 levels. Taken together, these findings demonstrate that Comm promotes midline crossing in the nerve cord by facilitating Robo1 ubiquitination by Nedd4, ultimately leading to its degradation.

## Introduction

In the embryo, neurons must find and form connections with the appropriate synaptic partners as they wire circuits in the developing nervous system. To make these connections, axons must travel long distances along complex but stereotyped trajectories, directed by a variety of secreted and cell-surface molecules that are conserved from invertebrates to humans. Interaction between these ligands and receptors on the membranes of axonal growth cones generates attractive and repulsive responses by triggering downstream changes in growth cone membrane and cytoskeletal architecture. Responses to these attractive and repulsive cues must be precisely coordinated to allow axons to grow along the appropriate paths and to find their proper post-synaptic targets.

Bilateral organisms must form neural circuits that coordinate the left and right halves of the body, a process that is essential for normal sensory, motor, and cognitive function. Defects in midline circuit

assembly can result in a variety of neurodevelopmental disorders (for reviews see *Engle, 2010*; *Izzi and Charron, 2011*) including horizontal gaze palsy with progressive scoliosis (*Bosley et al., 2005*), congenital mirror movement disorder (*Depienne et al., 2011*; *Srour et al., 2010*), autism spectrum disorder (*Anitha et al., 2008*; *Suda et al., 2011*), and agenesis of the corpus callosum (*Jamuar et al., 2017*; *Paul et al., 2007*). A subset of neurons in the CNS, called commissural neurons, contribute to the construction of these circuits by extending their axons across the embryonic midline to connect with targets on the opposite side of the body. Midline crossing requires a tight balance of attractive and repulsive responses (*Evans and Bashaw, 2010*; *Gorla and Bashaw, 2020*; *Nawabi and Castellani, 2011*). Pre-crossing commissural axons are initially drawn to the midline by several attractive ligands including netrin (*Ishii et al., 1992*; *Mitchell et al., 1996*; *Serafini et al., 1996*; *Serafini et al., 1994*) and sonic hedgehog (*Charron et al., 2003*). Upon reaching the midline, however, they respond to repulsive molecules such as slits and secreted semaphorins (*Brose et al., 1999*; *Kidd et al., 1999*; *Zou et al., 2000*) which allow them to exit the midline and keep them restricted to the contralateral side of the body by preventing re-crossing. Midline glia secrete both attractive and repulsive molecules concurrently, so an axon's behavior during a particular moment in time relies on tight regulation of receptors at its growth cone surface.

Pre-crossing commissural axons must suppress premature responses to repulsive signals to allow entry into the midline. In both vertebrates and invertebrates, axons prevent premature sensitivity to the midline repellant Slit by downregulating surface levels of its receptor, Robo1, until they reach the midline (*Evans et al., 2015*; *Keleman et al., 2002*; *Kinoshita-Kawada et al., 2019*; *Sabatier et al., 2004*). In *Drosophila*, Robo1 surface expression is downregulated by the Commissureless (Comm) protein (*Keleman et al., 2002*; *Kidd et al., 1998*; *Tear et al., 1996*), which is absolutely required for midline crossing. Nerve cords of Comm mutant embryos exhibit a complete lack of midline crossing (*Tear et al., 1996*), while Comm overexpression decreases Robo1 levels and induces ectopic midline crossing events (*Kidd et al., 1998*). Based on experiments in mammalian and insect cell culture, Comm is thought to act by preventing newly synthesized Robo1 from reaching the growth cone membrane by binding to it and shunting it to late endosomes, presumably leading to its degradation (*Keleman et al., 2002*; *Keleman et al., 2005*).

While it is generally accepted that Comm prevents Robo1 from being expressed on the growth cone in pre-crossing commissural axons, there are conflicting reports of the mechanism of Comm function. Within a conserved region of the Comm cytoplasmic domain, there are two PY motifs (PPCY and LPSY; *Myat et al., 2002*). In a previous study, mutation of the LPSY motif rendered Comm unable to localize Robo1 to endosomes in cell culture, and unable to induce midline crossing in vivo (*Keleman et al., 2002*). PY motifs are known binding motifs for the WW-domain containing Nedd4 family of HECT ubiquitin ligases (*Kanelis et al., 2001*; *Kasanov et al., 2001*), suggesting that Comm may downregulate Robo1 via interactions with these ligases. There remain several unanswered questions, however, concerning the roles of PY motifs in Comm function and the mechanistic relationship between Comm, Robo1, and Nedd4 family members during midline crossing in the embryonic nerve cord.

Firstly, it is debated whether Comm must be ubiquitinated by Nedd4-family ligases to effectively downregulate Robo1, as two previous studies investigating this question generated contradictory results. The first study demonstrated that Nedd4 ubiquitinated Comm, and that Comm required both its PY motifs and its cytoplasmic ubiquitin-accepting lysines to direct Robo1 to endosomes and downregulate its surface expression in cell culture (*Myat et al., 2002*). Interestingly, in the second study, an un-ubiquitinatable Comm mutant (Comm KR with all ubiquitin-accepting lysines mutated to arginines) was undistinguishable from wild-type Comm in its ability to localize itself and Robo1 to endosomes in vitro and induce Comm gain-of-function phenotypes in vivo (*Keleman et al., 2005*). In addition, it remains unknown whether Nedd4-family ligases are involved in midline crossing. When stained with a pan-neuronal marker, embryos homozygous for a chromosomal deletion that removes Nedd-4 had no noticeable defects in commissural axon guidance (*Keleman et al., 2005*). This finding alone, however, is insufficient to rule out a role for Nedd4 in midline crossing due to the possibility that loss-of-function effects could be masked by maternal deposition of Nedd-4, and/or possible redundancy with the other *Drosophila* Nedd4-family ligases Smurf and Su(dx).

In this study, we characterize the roles of PY motifs in Comm function and elucidate the mechanistic relationship between Comm, Robo1, and Nedd4-family ligases during midline crossing. To do this, we generated variants of Comm in which the LPSY motif (1PY) or both PY motifs (2PY) are mutated

and examined the effects of PY motif disruption on various Comm functions. In contrast to previous literature, we find that mutating the Comm LPSY motif alone does not completely disrupt proper localization to late endosomes in vitro and in vivo, direction of Robo1 away from the cell surface, and promotion of midline crossing. In addition, while previous studies only assayed the ubiquitination of Comm itself, we demonstrate that PY motifs are necessary for Comm to facilitate Robo1 ubiquitination and lysosomal degradation. Due to the importance of PY motifs for Comm function, we also investigated whether any of the three *Drosophila* Nedd4-family ligases play a role in midline crossing. Using sensitized genetic backgrounds, we provide evidence that loss of Nedd4, but not Smurf or Su(dx), lead to midline crossing defects in the embryonic nerve cord. In addition, we observe that Comm, Robo1, and Nedd4 interact as a three-member complex and that Comm PY motifs are necessary for Nedd4 incorporation into this complex. Finally, we demonstrate that exogenous Nedd4 enhances the ability of Comm to downregulate Robo1 protein levels in vivo. Taken together, these findings support a model in which Comm acts as a substrate adaptor to bring Robo1 into proximity to Nedd4, which it recruits via its PY motifs. This adaptor activity then allows Nedd4 to ubiquitinate Robo1, shunting it away from the growth cone membrane and to late endosomes and lysosomes for degradation. Finally, the resulting downregulation of Robo1 surface levels renders commissural axons insensitive to repulsive slit signals, allowing for midline crossing. In addition to providing new insights into the mechanism of Comm function, the work in this study reconciles two previously conflicting lines of research. Both earlier studies of Comm identified important aspects of Comm function, and by investigating key features of the conflicting models, we were able to unify them into a single coherent picture.

## Results
### The PY motifs of comm are required to promote midline crossing in vivo

As a first step to investigate the relationship between Comm and Nedd4-family ubiquitin ligases, we tested the roles of two putative Nedd4-family ligase binding sites in Comm's cytoplasmic tail: the PY motifs PPCY at amino acid positions 220–223 and LPSY at amino acid positions 229–232. Previous studies have demonstrated the necessity of these PY motifs for Comm's ability to promote midline crossing in vivo (*Keleman et al., 2002*; *Myat et al., 2002*); however, the previous studies used random insertions of Comm transgenes, making it very difficult to directly compare the in vivo activities of different mutant variants of the Comm protein.

To circumvent this issue, we generated transgenic flies carrying UAS transgenes of the following three Comm variants: WT (Wild-type), 1PY (LPSY->AASY), and 2PY (PPCY->AACY, LPSY AASY, *Figure 1A*), and inserted them into the same integration site on the third chromosome to ensure that these transgenic flies expressed equivalent levels of Comm transcripts when crossed to flies bearing neuronal gal4 drivers. To allow us to vary the expression levels of these different Comm variants, we created two sets of transgenic flies. In the first set, we cloned the Comm coding sequence downstream of 5 copies of the UAS promoter sequence (5 X UAS) and in the second set, we used 10 copies of the UAS sequence (10 X UAS). We then assayed the ability of the variants to induce misrouting of ipsilateral axons across the midline in the embryonic nerve cord. In these assays, we examine the eight posterior-most segments of the nerve cord, which are each comprised of an anterior and posterior commissure or 16 total commissures. First, we expressed these 5 X UAS variants under the pan-neuronal Elav-gal4 driver, and investigated their effect on FasII neurons, which run in three parallel ipsilateral bundles along the longitudinal tracts of the nerve cord (*Figure 1B*). We found that WT Comm induced ectopic FAS II crossing in roughly half the commissures in the nerve cord (mean = 52%, n=17). In contrast to previously published data, Comm 1PY was also able to induce ectopic FAS II crossing, although it was significantly less effective than WT (p<0.0001, ANOVA/Tukey, n=23). Finally, nerve cords of embryos expressing Comm 2PY were indistinguishable from those of control embryos not carrying any Comm transgenes (p=0.997, ANOVA/Tukey, 2PY n=20, CTRL n=18; *Figure 1B and C*).

To investigate the ability of Comm to induce ectopic midline crossing in a cell-autonomous manner, we expressed 5 X UAS Comm variants in apterous neurons using the apGal4 driver and investigated their effect on this specific ipsilateral neuron population. The apGal4 element is expressed in only three neurons per hemisegment allowing for greater resolution of midline crossing phenotypes. We

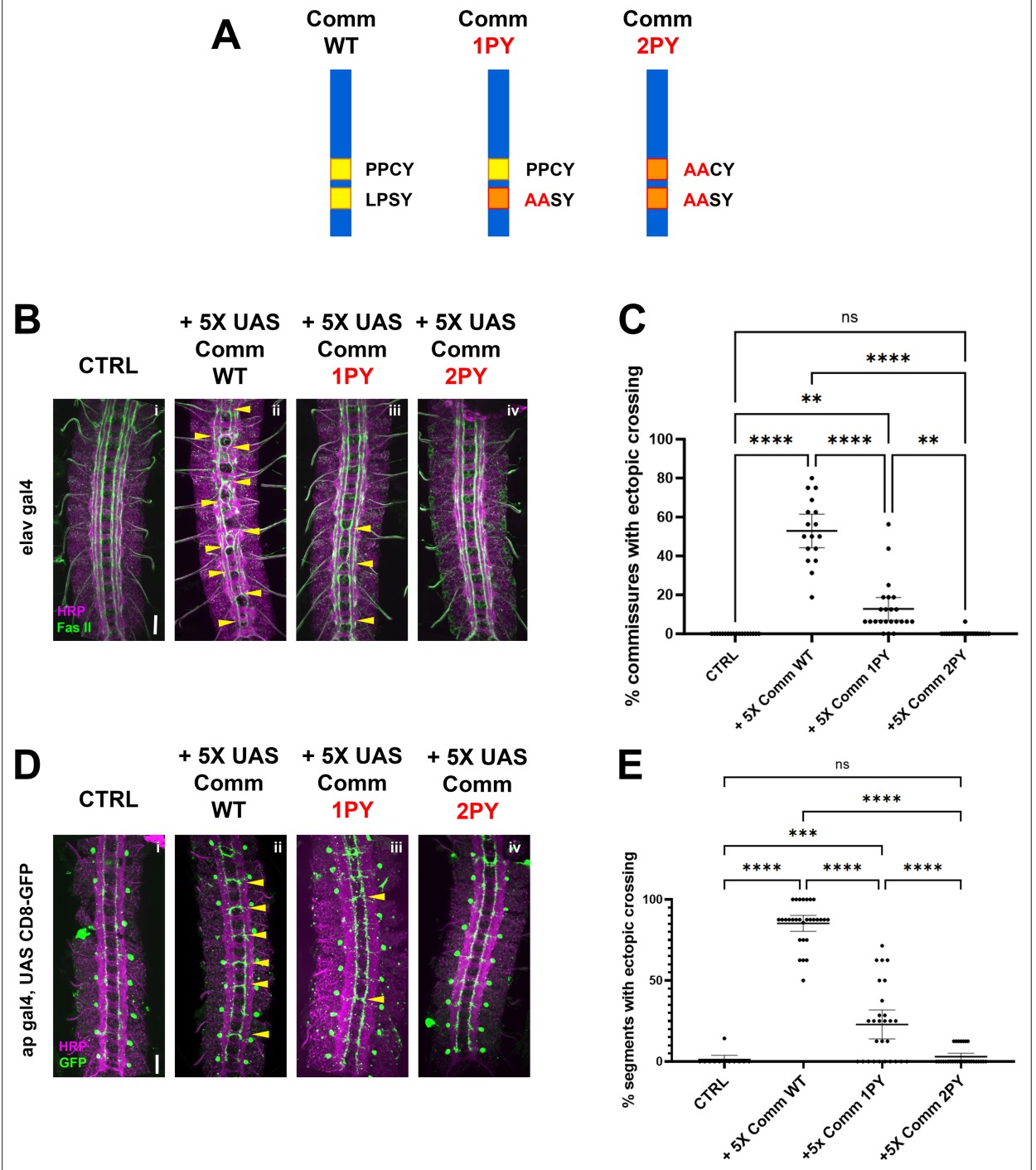

**Figure 1.** PY motifs are required for Comm to promote midline crossing. (**A**) Comm variants used in this study including WT, and mutant forms in which the LPSY (1PY) or both the LPSY and PPCY (2PY) motifs are disrupted. (**B–C**) PY motifs are necessary for Comm to cause ectopic crossing in the ipsilateral FasII neuron population when driven pan-neuronally. (**B**) Micrographs of stage 16–17 nerve cords stained with HRP (pan-neuronal marker, magenta) and FasII (green) expressing no Comm (**i**), Comm WT (**ii**), Comm 1PY (**iii**), and Comm 2PY (**iv**) under the elav gal4 driver. (**C**) Quantification of ectopic FasII crossing in stage 16–17 embryos expressing wild type and PY-mutant Comm variants. Groups were compared using ANOVA with Tukey's post-hoc test, and error bars represent mean with 95% confidence interval. (**D–E**) PY motifs are required for Comm to cell-autonomously promote ectopic midline crossing when driven in apterous neurons. (**D**) Micrographs of stage 16–17 nerve cords stained with HRP (magenta) and GFP (green), expressing GFP and no Comm (**i**), Comm WT (**ii**), Comm 1PY (**iii**), and Comm 2PY (**iv**) under the ap gal4 driver. (**E**) Quantification of ectopic apterous neuron crossing in stage 16–17 embryos expressing wild type and PY-mutant Comm variants. Groups were compared using ANOVA with Tukey's post-hoc test, and

*Figure 1 continued on next page*

*Figure 1 continued*

error bars represent mean with 95% confidence interval. For all graphs, each data point represents one embryo. ** (p<0.01) **** (p<0.0001). Scale bars represent 20 μM.

The online version of this article includes the following source data and figure supplement(s) for figure 1:

**Source data 1.** This is the raw scoring data for the phenotypes presented.

**Source data 2.** This is the raw scoring data for the phenotypes presented.

**Source data 3.** This is the raw scoring data for the phenotypes presented.

**Source data 4.** This is the raw scoring data for the phenotypes presented.

**Source data 5.** This is the raw scoring data for the phenotypes presented.

**Source data 6.** This is the raw scoring data for the phenotypes presented.

**Figure supplement 1.** Driving high levels of Comm, including those lacking functional PY motifs, induces ectopic midline crossing in ipsilateral neurons.

observed that Comm expression induces ectopic apterous crossing in the anterior commissure of nerve cord segments. Nearly all nerve cord segments of embryos expressing WT Comm had ectopic apterous crossing events (mean = 85.28%, n=29). Consistent with our FasII data, Comm 1PY induced significantly fewer ectopic crossing events than WT, and Comm 2PY was completely unable to induce ectopic crossing (*Figure 1D and E*).

As expected, pan-neuronally expressing higher levels of WT Comm using 10X-UAS results in a much more severe ectopic crossing phenotype; indeed, these embryos have a *slit* mutant phenotype (*Figure 1—figure supplement 1*). Surprisingly, we observed that pan-neural expression of 10 X Comm-2PY, which has no effect when expressed at lower levels, leads to a striking gain of function phenotype where many segments display strong ectopic crossing, both with FasII and with HRP (*Figure 1—figure supplement 1*). Taken together, these data indicate that intact PY motifs play an important and a dose-dependent role in Comm's ability to induce midline crossing and raise the possibility that Comm may also inhibit midline repulsion independently of its PY motifs.

## Comm PY motifs are required for Robo1 ubiquitination and lysosomal degradation

In vivo, Comm overexpression has been shown to reduce Robo1 protein levels and produces a phenotype characterized by excessive midline crossing, reminiscent of the Robo1 loss-of-function phenotype (*Kidd et al., 1998*). However, previous studies have not investigated the mechanism underlying Robo1 protein degradation. To determine the role of Comm PY motifs in reducing Robo1 protein levels and to define the degradative mechanism, we co-expressed Robo1 and Comm variants in S2*R* + cells and measured the expression of both Comm and Robo1 by quantitative Western blotting. We found that co-expression of WT comm significantly reduces Robo1 levels compared to those in cells transfected with Robo1 alone (*Figure 2A and B*). Comm 1PY also reduces Robo1 levels, though it is significantly less effective at doing so than WT Comm, while Comm 2PY is completely unable to reduce Robo1 levels (*Figure 2A and B*). Interestingly, mutating the PY motifs in Comm also affects stability of the Comm protein itself. Levels of Comm 1PY and 2PY are elevated relative to WT Comm (*Figure 2A and C*). Taken together, these findings suggest that PY motifs are critical for turnover of the Comm protein, as well as its ability to facilitate Robo1 degradation.

Since PY motifs can act as binding sites for Nedd4-family ubiquitin ligases (*Kanelis et al., 2006*; *Kanelis et al., 2001*; *Kasanov et al., 2001*), we next investigated whether ubiquitination plays a role in regulating Robo1 and Comm protein stability. Previous studies demonstrated that Comm is ubiquitinated (*Myat et al., 2002*), but this ubiquitination appears to be dispensable for its function including proper localization, and promotion of midline crossing, since mutation of all of Comm's cytoplasmic lysines (Comm KR) has no apparent effect on its localization or function (*Keleman et al., 2005*). Somewhat surprisingly, the possibility that Comm facilitates Robo1 ubiquitination was never investigated. This led us to evaluate the possibility that Comm downregulates Robo1 levels by facilitating Robo1 ubiquitination and therefore, leading to its degradation. To test whether Comm facilitates Robo ubiquitination, we co-transfected S2*R* + cells with Robo1 and the Comm variants along with a construct expressing Flag-tagged ubiquitin, and then immunoprecipitated Robo1 and assessed its ubiquitination levels (*Figure 2D and E*). WT Comm significantly increases Robo1 ubiquitination

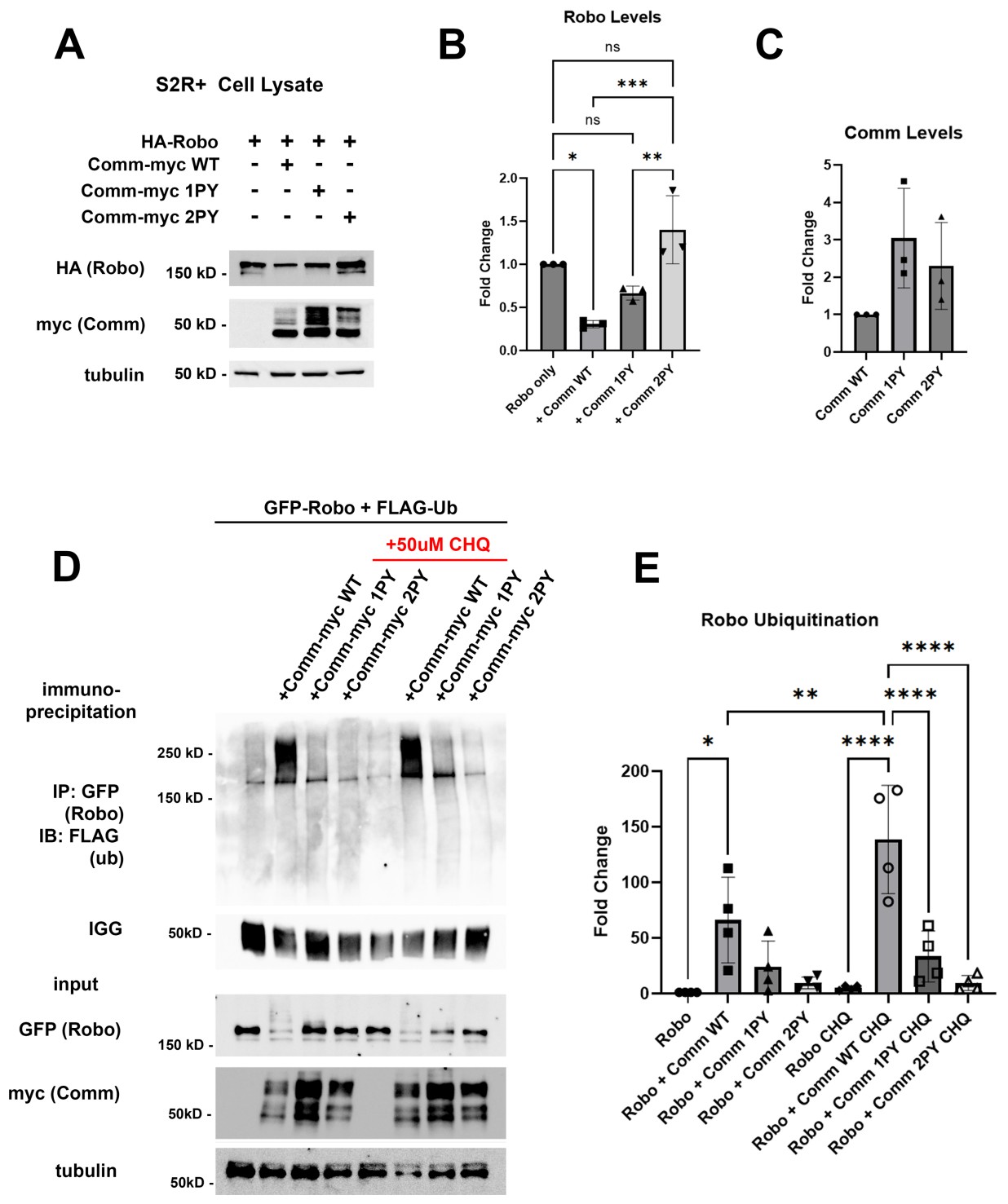

**Figure 2.** PY motifs are required for Comm to downregulate Robo1 protein levels in vitro. (**A–C**) Loss of PY motifs inhibits Robo1 degradation facilitated by Comm and increases Comm stability. (**A**) Western blot of S2*R* + cells transfected with HA-Robo1 and Comm-Myc WT, 1PY, or 2PY. (**B**) Fold change in Robo1 levels relative to the 'Robo1 alone' condition. Protein levels were calculated via densitometry and normalized to tubulin. (**C**) Fold change in Comm levels relative to the 'WT Comm' condition. Protein levels were calculated via densitometry and normalized to tubulin. For B and C, groups were compared using ANOVA with Tukey's post hoc test. Error bars represent mean +/-SD. Each data point represents values calculated from a single experiment. * (p<0.05), ** (p<0.01), *** (p<0.001). (**D–E**) Comm requires PY motifs to facilitate Robo1 ubiquitination and degradation in the lysosome. (**D**) Immunoprecipitation of ubiquitinated Robo1 in control or chloroquine-treated S2*R* + cells transfected with FLAG-Ubiquitin and HA-Robo1 alone, or HA-Robo1 and WT, 1PY, or 2PY Comm-myc variants. Cells were incubated with Chloroquine for 2 hr. (**E**) Fold change of ubiquitinated Robo1 levels

*Figure 2 continued on next page*

*Figure 2 continued*

relative to the 'Robo1 only' condition. Ubiquitinated Robo1 levels were measured by quantifying immunoprecipitated FLAG blotting via densitometry and normalizing to IGG, lysate Robo1, and tubulin. For B and C, groups were compared using ANOVA with Tukey's post hoc test. Error bars represent mean +/-SD. Each data point represents values calculated from a single experiment * (p<0.05), ** (p<0.01), **** (p<0.0001).

The online version of this article includes the following source data for figure 2:

**Source data 1.** This zip file contains raw western blot images and requested files.

**Source data 2.** This zip file contains raw western blot images and requested files.

**Source data 3.** This zip file contains raw western blot images and requested files.

**Source data 4.** This zip file contains raw western blot images and requested files.

**Source data 5.** This is a spreadsheet containing quantitative data for *Figure 2*.

**Source data 6.** This is a spreadsheet containing quantitative data for *Figure 2*.

levels relative to those observed in cells transfected with Robo1 alone and removing either LPSY or both PY motifs eliminates Comm's ability to enhance Robo1 ubiquitination (*Figure 2D and E*). Importantly, in cells co-transfected with Robo1 and WT Comm, treatment with the lysosomal inhibitor chloroquine significantly stabilizes ubiquitinated Robo1 (*Figure 2D and E*). Taken together, our results show that Comm facilitates Robo1 ubiquitination in a PY motif-dependent manner and this ubiquitinated Robo1 is degraded in the lysosome.

To test the effects of expressing Comm on Robo protein levels in vivo, we measured the expression of endogenous Robo1 protein in embryos over-expressing 5 X UAS Comm transgenes by quantitative immunofluorescence (*Figure 3*). In stage 15–16 embryos, endogenous Robo1 is expressed at high levels in the longitudinal portions of the axon scaffold and is largely absent from the commissural portions of projecting axons (*Figure 3Ai*). Overexpression of WT comm under the elavGal4 driver significantly reduces endogenous Robo1 levels (*Figure 3Aii*, B). As predicted from our in vitro data, loss of one PY motif significantly diminished Comm's ability to downregulate Robo1 levels, while loss of both eliminated the ability altogether and endogenous Robo1 levels are indistinguishable from control embryos (*Figure 3Aii*, B). Since expression of high levels of Comm 2PY with elavGal4 (10x-UAS-*Figure 1—figure supplement 1*) is still able to induce significant ectopic midline crossing, we were curious whether endogenous Robo1 levels are also reduced in embryos over expressing 10X-UAS Comm 2PY. Intriguingly, while expression of either 10 X UAS Comm WT or Comm 1PY leads to even stronger reduction in endogenous Robo1 than the 5 X transgenes, expression of 10 X UAS Comm 2PY does not reduce endogenous Robo1 levels at all (*Figure 3—figure supplement 1A and B*). We also tested the effects of mutating the PY motifs on the expression of the Comm protein itself, and consistent with our in vitro experiments, we find that loss of PY motifs increases Comm levels in a dose-dependent manner (*Figure 3A and C*). Interestingly, even when over-expressed using the Gal4-UAS system, Comm (both 5 X and 10 X UAS) is barely detected unless the PY motifs are mutated. Taken together, these data suggest that PY motifs control the stability of the Comm protein, as well as its ability to facilitate Robo1 degradation. In addition, our observation that expression of 10 X UAS Comm 2PY does not reduce endogenous Robo1 levels suggests that in the absence of its PY motifs, Comm can still inhibit midline repulsion and that this effect is not due to reduced levels of Robo1 protein (see Discussion).

## The PY motifs of comm act cooperatively to regulate comm and Robo1 localization

As Comm has been proposed to promote midline crossing by preventing Robo1 from reaching the growth cone surface (*Keleman et al., 2002*; *Keleman et al., 2005*), we next investigated the involvement of PY motifs in the intracellular localization of both Comm and Robo1. Previous studies used mammalian COS-7 cells to examine Comm's effect on Robo1 localization because of their large size and spread morphology (*Keleman et al., 2002*; *Keleman et al., 2005*); thus, to allow comparison with the earlier studies, we first co-transfected COS-7 cells with HA-Robo1 and Myc-Comm variants and performed immunostaining for each to visualize their localization. In cells transfected with Robo1 alone, Robo1 is diffusely localized throughout the cell, in intracellular puncta, and at the cell surface (*Figure 4A*). As observed in previous studies (*Keleman et al., 2002*; *Keleman et al., 2005*; *Myat et al., 2002*), when co-transfected with WT Comm, we see a dramatic change in Robo1 localization.

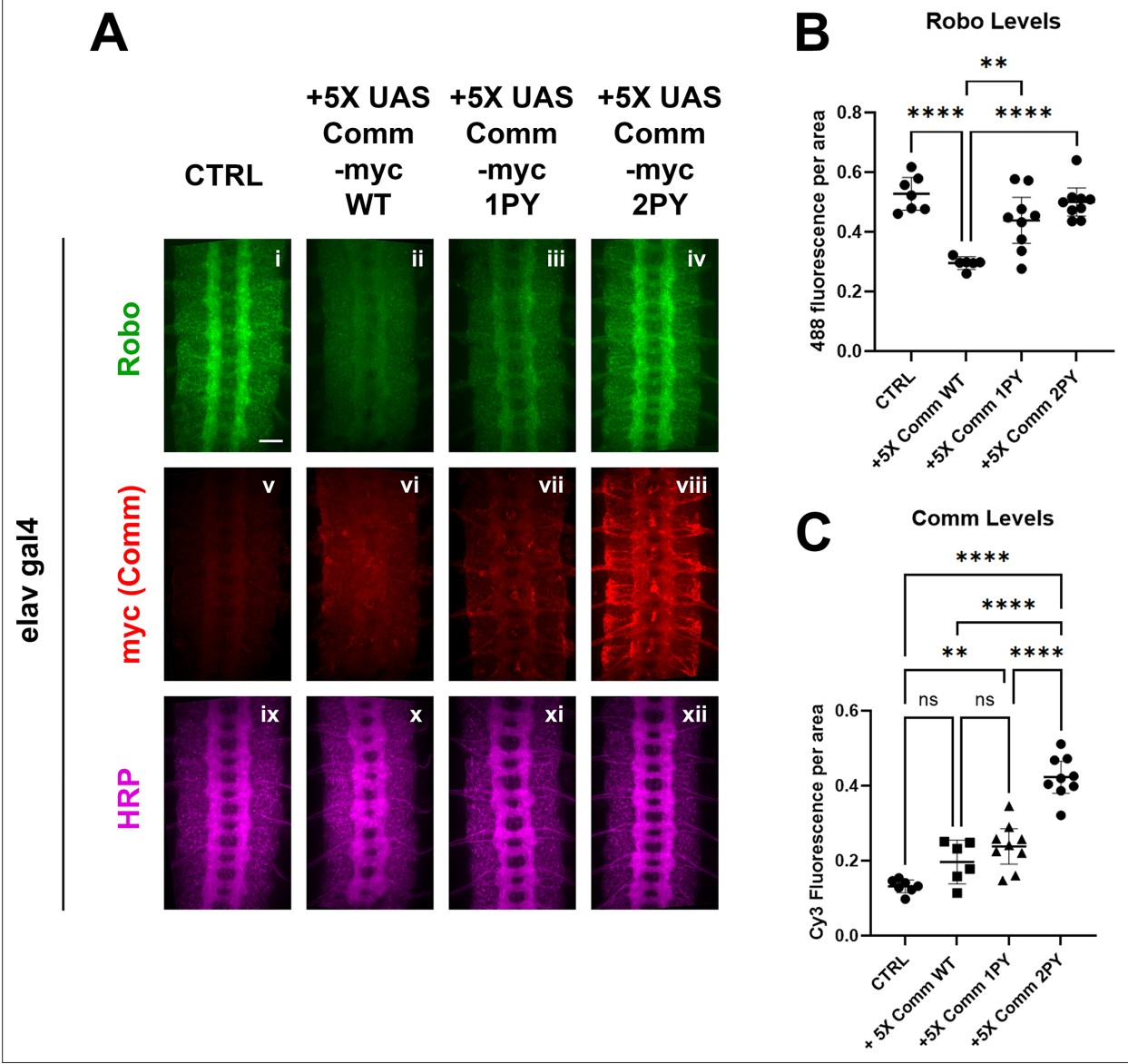

**Figure 3.** PY motifs are necessary for Comm to downregulate Robo protein levels in vivo. (**A**) Stage 15–16 embryos expressing no Comm or 5 X UAS Comm-myc variants under the pan-neuronal elav gal4 driver. Embryos are stained for endogenous Robo1 (**i–iv**), Myc (v-viii), and pan-neuronal marker HRP (ix-xii). Scale bar represents 20 μM. (**B**) Quantification of endogenous Robo1 levels in stage 15–16 embryos expressing Comm variants under elav gal4 driver. Robo1 levels were measured by creating a mask of the axonal scaffold, measuring 488 fluorescence within the scaffold, and dividing by scaffold area. (**C**) Quantification of Comm levels in stage 15–16 embryos expressing Comm variants under the *elavGal4* driver. Comm levels were measured by creating a mask of the whole nerve cord, measuring Cy3 fluorescence within the nerve cord, and dividing by nerve cord area. For B and C, groups were compared using ANOVA with Tukey's post-hoc test. ** p<0.01 **** p<0.0001. Error bars represent 95% confidence intervals around the mean. Each data point represents one embryo.

The online version of this article includes the following source data and figure supplement(s) for figure 3:

**Source data 1.** This is the raw quantitative date used to generate the graphs in *Figure 3*.

**Figure supplement 1.** Robo expression in 10XUAS Comm transgenic flies.

**Figure supplement 1—source data 1.** This is a spreadsheet containing raw data for the graphs in *Figure 3—figure supplement 1*.

Instead of diffuse localization throughout the cell, Robo1 is predominantly found with Comm in intracellular puncta (*Figure 4A and B*). In contrast to previous findings (*Keleman et al., 2002*), disruption of the LPSY motif alone (Comm 1PY) does not alter Comm localization compared to WT; however, Comm 1PY is significantly less effective at routing Robo1 to intracellular puncta, and we observe a return of diffuse Robo1 expression (*Figure 4A–C*). In contrast to Comm 1PY, Comm2PY shows diffuse

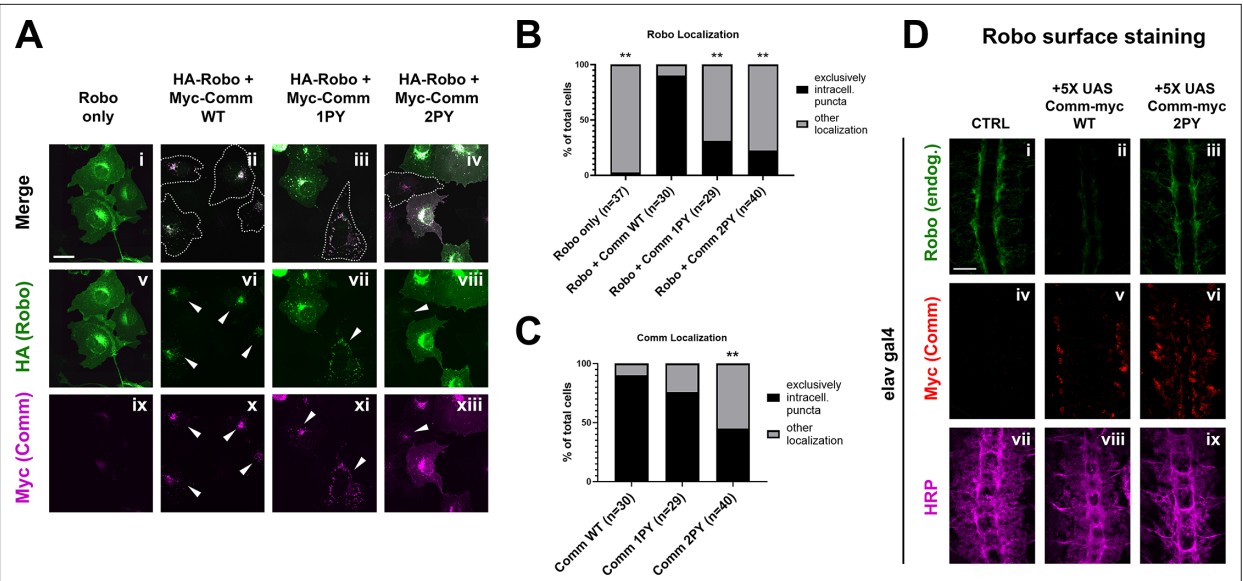

**Figure 4.** PY motifs are necessary for Comm to effectively traffic Robo1. (**A**) COS-7 cells transfected with HA-Robo1 and Myc-Comm variants. Cells in which Comm and Robo1 are exclusively in puncta are outlined with dotted lines in merged images and labelled with arrowheads in images of individual color channels. Scale bar represents 15 µM. (**B**) Quantification of cells expressing Robo1 exclusively in intracellular puncta or in diffuse localization throughout the cell. N represents the number of individual cells counted, which were taken from seven images of cell fields chosen at random locations on the microscope slide. Distribution of exclusively punctate vs diffuse Robo1 localization in Comm-transfected conditions was compared to the Robo1 alone condition via Chi Square (** p<0.01). (**C**) Quantification of cells expressing Comm exclusively in intracellular puncta or in diffuse localization throughout the cell. N represents the number of individual cells counted, which were taken from seven images of cell fields chosen at random locations on the microscope slide. Distribution of exclusively punctate vs diffuse Comm localization in comm mutant conditions was compared to the WT Comm condition via Chi Square (** p<0.01). In both B and C, only cells expressing both Comm and Robo1 were counted. (**D**) Surface expression of Robo in nerve cords of stage 14–15 embryos. Embryos were live-dissected, left unpermeabilized, and stained with an antibody against endogenous Robo1 at 4 °C. After Robo1 staining, embryos were permeabilized and stained with HRP and Myc. Scale bar represents 20 µM.

The online version of this article includes the following source data for figure 4:

**Source data 1.** This is the raw data from the cell localization counts presented in *Figure 4*.

localization throughout the cell, as well as impaired ability to route Robo1 to intracellular puncta. In summary, the LPSY motif is dispensable for proper Comm localization in cell culture but is required for effective routing of Robo1, while loss of both PY motifs, causes Comm mis-localization as well as a failure to recruit Robo1 to intracellular puncta.

To explore the ability of Comm to regulate the surface expression of Robo1 and to further evaluate the importance of the PY motifs in this context, we performed a series of experiments in live dissected unpermeabilized stage 15 embryos. This method allows us to detect the amount of endogenous Robo1 that is present on the axonal surface in various genetic backgrounds. In control embryos, Robo1 is present on the surface of longitudinal portions of axons in the developing ventral nerve cord (*Figure 4D*). In contrast, minimal Robo1 is detected on the axonal surface in embryos expressing 5 X UAS wild-type Comm (*Figure 4D*). In embryos expressing 5 X UAS Comm 2PY, Robo1 is expressed normally on the axonal surface. These observations support our localization studies in COS-7 cells and suggest that Comm can prevent Robo1 surface expression in vivo.

## Comm can recruit Robo1 to late endosomes in vivo

To further investigate the role of PY motifs in Comm localization in vivo, and to define the subcellular localization of Comm, we ectopically expressed UAS transgenes of the Comm variants using the apGal4 driver. Although ap neurons are ipsilateral and do not express endogenous Comm, they are a small and sufficiently sparse neuronal population to allow for easy visualization of both axons and cell bodies. We found that 10 X UAS WT Comm is restricted to cell bodies and displays a punctate expression pattern indicative of localization in intracellular vesicles (*Figure 5A and B*). Like WT Comm, 10 X UAS Comm 1PY is observed in cell body puncta. However, Comm 1PY, unlike WT Comm, is also

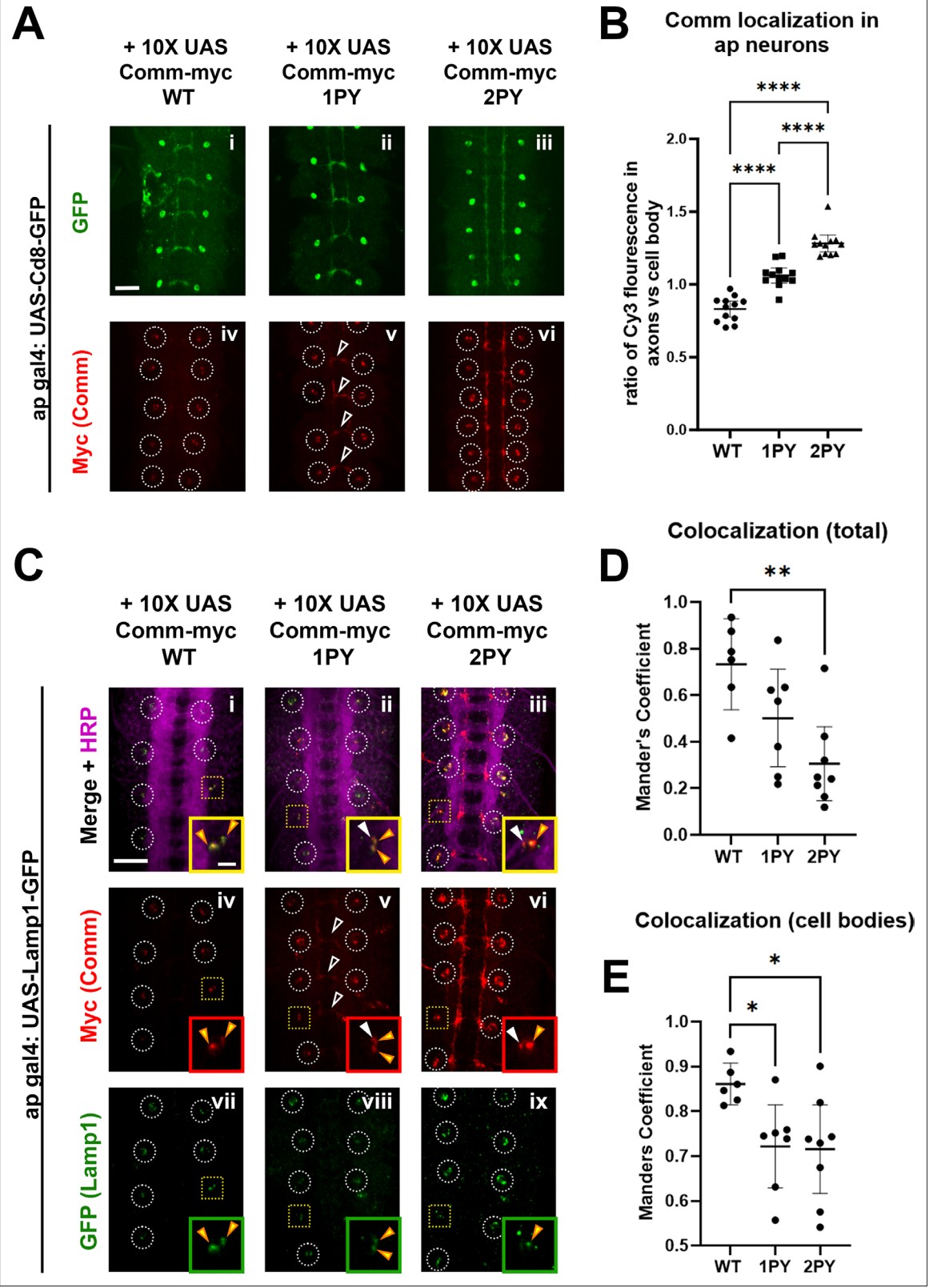

**Figure 5.** Comm requires intact PY motifs to localize appropriately in vivo. (**A**) Micrographs of stage 16–17 embryos expressing CD8-GFP and WT or PY-mutant Comm-myc variants under the apgal4 driver. Embryos are stained with GFP to visualize apterous neurons, Myc (Comm), and HRP (pan-neuronal marker). Cell bodies are circled in panels showing Myc staining (iv-vi). In v, axonal Comm expression is indicated with arrowheads. Scale bar represents 20 μM. (**B**) Quantification of the ratio of axonal Comm to cell soma Comm in stage 17 embryos expressing WT and PY-mutant Comm variants

*Figure 5 continued on next page*

*Figure 5 continued*

in apterous neurons. Axonal and cell soma comm were measured within masks created of these two features, derived from images of apterous neurons in the GFP channel. Axon/cell soma ratios were compared via ANOVA with Tukey's post-hoc test (**** $p<0.0001$). Error bars represent 95% confidence intervals around the mean. Each data point represents a single embryo. (**C**) Stage 15 embryos expressing Lamp1-GFP and WT or PY-mutant Comm myc variants under the *apGal4* driver. Embryos are stained with GFP (Lamp1), Myc (Comm), and HRP (pan-neuronal marker). Cell bodies are outlined with white circles except those that are enlarged in inset images, which are outlined with yellow squares. Within inset images, Comm/Lamp1 co-positive puncta are indicated with yellow arrowheads and puncta containing Comm alone are indicated with white arrowheads. Scale bar represents 20 µM in the large image and 5 µM in the inset image. (**D**) Total colocalization of Comm and Lamp1. The proportion of areas of Comm expression that were also positive for Lamp1 was calculated via pearson's coefficient within a mask created of the area of Comm expression using smoothened images from the Comm channel. (**E**) Colocalization of Comm and Lamp1 within cell bodies. The proportion of areas of Comm expression that were also positive for Lamp1 was calculated via Pearson's coefficient within a mask created of cell bodies using smoothened images from the Lamp1 channel. For D and E, colocalization across conditions was compared via ANOVA with Tukey's post hoc test (* $p<0.05$, ** $p<0.01$). Error bars represent 95% confidence intervals around the mean. Each data point represents one embryo.

The online version of this article includes the following source data and figure supplement(s) for figure 5:

**Source data 1.** This is the raw data used to produce the localization graphs in *Figure 5*.

**Source data 2.** This is the raw data used to produce the localization graphs in *Figure 5*.

**Source data 3.** This is the raw data used to produce the localization graphs in *Figure 5*.

**Figure supplement 1.** Robo colocalization with Comm is unaffected by mutations in Comm PY motifs.

**Figure supplement 1—source data 1.** This is the raw data used to produce the localization graphs in *Figure 5*, *Figure 5—figure supplement 1B*.

**Figure supplement 1—source data 2.** This is the raw data used to produce the localization graphs in *Figure 5*, *Figure 5—figure supplement 1C*.

**Figure supplement 2.** Colocalization of Comm with the late endosomal marker Rab 7.

present in axons. 10 X UAS Comm 2PY displays even greater enrichment in axons (*Figure 5A and B*). When Robo1 is co-expressed with 10 X UAS Comm, its expression pattern mimics that of the Comm variant with which it is expressed (*Figure 5—figure supplement 1*). Notably, when co-expressed with WT Comm, Robo1 expression is absent from axons and restricted to the cell body, often in Comm-positive intracellular puncta. Meanwhile, Robo1 co-expressed with PY mutant Comm variants can be found alongside these Comm variants in axons (*Figure 5—figure supplement 1A*). Robo co-localizes equally with WT and PY mutant Comm variants in vivo (*Figure 5—figure supplement 1B and C*), suggesting that Robo1 may get 'dragged' out of the cell body by Comm variants that are unable to localize correctly. The observation that mutating Comm PY motifs leads to 'leakage' of both Comm and Robo1 into axons, coupled with a failure of these mutants to down-regulate Robo1 and promote midline crossing strongly suggest that the primary site of Comm function is in the cell soma. As most actively degradative intracellular compartments are restricted to the soma in neurons (for a review of the neuronal endolysosomal system, see *Roney et al., 2022*), these findings support the role of PY motifs for routing Comm and its cargo to late endosomes and lysosomes.

To determine whether Comm localizes to late endosomes and lysosomes, we co-expressed 10 X UAS Comm variants with GFP-tagged late endosomal/lysosomal markers using apGal4. Two such markers, Rab7 and Lamp1, are observed in a punctate pattern within cell bodies but, as expected, are largely excluded from axons when overexpressed in apterous neurons (*Figure 5C-E*, *Figure 5—figure supplements 1 and 2*). Each Comm variant can be observed in puncta positive for Rab7 or Lamp1 (*Figure 5*, *Figure 5—figure supplement 2*); however, both Comm PY mutant variants show a significant reduction in total co-localization with these late endosomal and lysosomal markers relative to wild-type Comm (*Figure 5C–E*, $p<0.05$ for both 1PY and 2PY, ANOVA). This reduction is largely due to the fact that axonally localized Comm 1PY or 2PY does not co-localize with either marker (*Figure 5C-E*, *Figure 5—figure supplement 2*). Interestingly, PY mutant Comm variants show significantly less co-localization with Lamp1 even when only observing Comm present in cell bodies, suggesting that with Lamp1, total co-localization is not only decreased due to Comm leakage into axons, but also due to compromised routing to the correct subcellular compartments within cell bodies (*Figure 5C and E*). As Robo localization largely follows that of the Comm variant with which it is co-expressed (*Figure 5*, *Figure 5—figure supplement 1*), its routing to degradative compartments is likely also compromised when co-expressed with PY mutant Comm variants. Collectively, these findings demonstrate that Comm localizes to late endosomes and lysosomes in vivo and that PY motifs keep Comm restricted to these compartments. In summary, PY motifs are critical for proper

localization of Comm to degradative compartments and its ability to route its cargo Robo1 away from the axonal cell surface.

## Nedd4 promotes midline crossing of commissural axons

In the experiments described, we have demonstrated the importance of PY motifs for numerous Comm functional outputs: promotion of midline crossing, targeting to endosomes/lysosomes, Robo1 trafficking, and promotion of Robo1 ubiquitination and degradation. As these motifs are known to bind Nedd4-family ubiquitin ligases and are indispensable for Comm function, we investigated whether any of the Nedd4-family ligases play a role in midline crossing in vivo. Previous studies have led to conflicting conclusions about the importance of Nedd4 for midline crossing. One study found that Nedd4 overexpression increases the Comm gain-of-function phenotype characterized by ectopic midline crossing *Myat et al., 2002*; however, a different research group was unable to replicate these results and also observed that embryos homozygous for a chromosomal deletion covering the Nedd4 locus show no obvious midline crossing defects (*Keleman et al., 2005*). These results call into question the role of Nedd4 in midline crossing. Given our biochemical results showing that Comm promotes Robo1 ubiquitination through its predicted Nedd4 binding sites and the potential redundancy among the three *Drosophila* Nedd4 family ligases, we revisited the role of Nedd4 ubiquitin ligases in commissural axon guidance and in the regulation of Robo1 using genetic and biochemical approaches.

To evaluate a potential role for Nedd4 in midline crossing, we first examined its expression pattern in developing embryos using a MiMIC insertion (*Nagarkar-Jaiswal et al., 2015*), which inserts a GFP tag into the coding sequence of the endogenous Nedd4 locus (Nedd4-GFP). Using this tool, we observe a strong enrichment of Nedd4 protein in the ventral nerve cord of stage 14 embryos, a stage when many commissural axons are extending across the midline. Using this line, we are able to detect Nedd4 protein in both cell bodies and CNS axons, indicating that Nedd4 is present at the right time and place to play a role in commissural axon guidance (*Figure 6—figure supplement 1*). To test for a functional role of Nedd4 in midline guidance, we used the well-characterized Nedd4T119FS allele, which was generated via EMS mutagenesis and induces a frameshift mutation at the 119th amino acid. This frameshift truncates the Nedd4 protein, such that it lacks its WW (PY-motif binding) domains and HECT ubiquitin ligase domain, which renders it non-functional (*Sakata et al., 2004*). When we examined embryos homozygous for this loss of function allele with HRP immunofluorescence, we detected no obvious defects (*Figure 6A*), consistent prior observations using a chromosomal deletion for Nedd4 (*Keleman et al., 2005*). Since Nedd4 transcript is known to be maternally deposited, we sought to further limit Nedd4 function by generating maternal/zygotic mutants. However, these approaches were not successful, presumably due to the necessity of Nedd4-family ligases in oogenesis (*Xia et al., 2010*) as well as early patterning events and cell fate decisions that occur before axon guidance begins (*Bakkers et al., 2005*; *Lin et al., 2017*; *Lohraseb et al., 2022*; *Persaud et al., 2011*; *Podos et al., 2001*; *Zhang et al., 2014*).

To more thoroughly examine if Nedd4 plays any role in commissural axon guidance, we used two sensitized genetic backgrounds that lead to partial disruptions in midline crossing, *frazzled* loss-of-function mutants and ectopic expression of a truncated Frazzled receptor, FraΔC (*Garbe et al., 2007*). We and others have used these genetic backgrounds to demonstrate important functional roles for a number of genes in commissural axon guidance that are otherwise undetectable, including Robo2 and the transmembrane and secreted Semaphorins (Sema1a and Sema2A, B, respectively; *Arbeille and Bashaw, 2018*; *Evans et al., 2015*; *Hernandez-Fleming et al., 2017*; *Spitzweck et al., 2010*). In *fra* mutants, there are significant disruptions in midline crossing that can be observed when visualizing the entire axon scaffold as well as when visualizing a subset of commissural axons (labeled by egGal4 expressing CD8-GFP). Specifically, *fra* mutants display thin or missing posterior commissures in approximately 30% of embryonic segments. In egGal4-expressing commissural neurons, the EW cluster of neurons normally project across the posterior commissure and approximately 35% of EW neurons fail to project their axons across the midline in *fra* mutants (*Figure 6A–D*). While removing one copy of Nedd4 does not increase crossing defects in the *fra*-null background, as measured by commissure thickness with HRP (*Figure 6A and B*) or EW crossing (*Figure 6C and D*), removing both copies of Nedd4 leads to a significant enhancement of the *fra* mutant phenotype, and a profound disruption in midline crossing, in some instances resembling *comm* loss of function mutants (*Figure 6A*). *fra, nedd4* double mutant embryos are missing over half of their commissures (compared to the <10% missing in

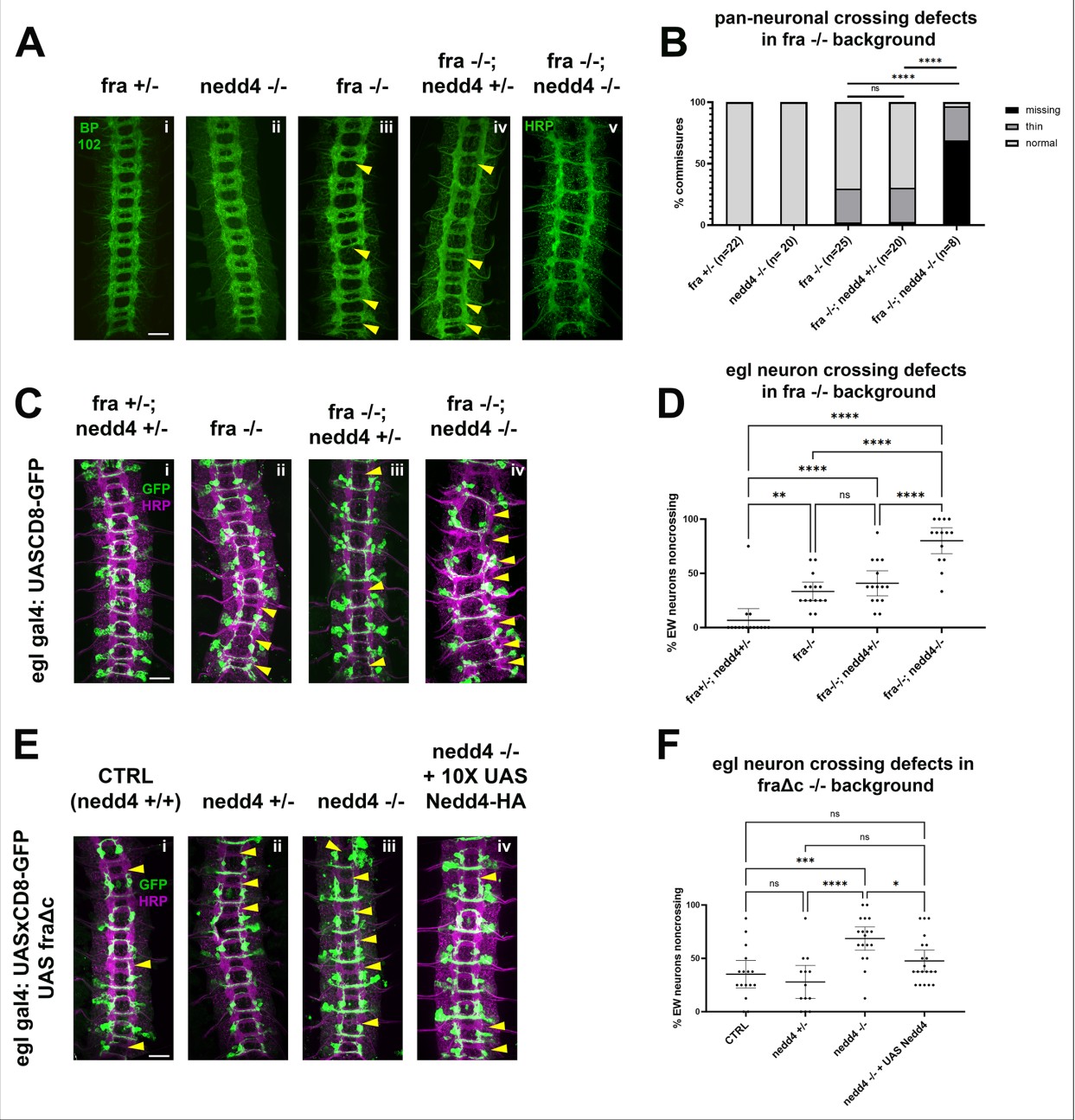

**Figure 6.** Nedd4 is required for midline crossing in vivo. (**A–B**) Loss of *nedd4* enhances pan-neuronal crossing defects in a *fra-/-* background. (**A**) Stage 15–16 embryos with various combinations of *wild-type* and null *fra* and *nedd4* alleles, stained with pan-neuronal markers BP102 (**i–iv**) and HRP (**v**). Commissures with crossing defects (either thin or completely missing) are indicated with yellow arrowheads. (**B**) Quantification of crossing defects in embryos within the genotypes indicated in A. Percentage of total crossing defects was compared across groups using ANOVA (** p<0.01, **** p<0.0001). N represents number of individual embryos. (**C–D**) Loss of *nedd4* enhances EW crossing defects in a *fra -/-* background. (**C**) Stage 15–16 embryos with various combinations of wild-type and null *fra* and *nedd4* alleles, expressing GFP under the egl gal4 driver. Embryos are stained with pan-neuronal marker HRP, and GFP to visualize eg neurons. Commissures with EW crossing defects are indicated with arrowheads. (**D**) Quantification of crossing defects in embryos within the genotypes indicated in (**C**). Percentage of crossing defects was compared across groups using ANOVA (** p<0.01, **** p<0.0001). Each data point represents an individual embryo and error bars represent 95% confidence intervals around the mean (**E–F**) Loss of *nedd4* enhances EW crossing defects in the FraΔc background. (**E**) Embryos with *nedd4* mutations and/or exogenous Nedd4 expression in the FraΔc background. (**F**) Quantification of EW crossing defects for the genotypes shown in (**E**). Percentage of crossing defects was compared across groups using ANOVA (** p<0.01, **** p<0.0001). Each data point represents an individual embryo and error bars represent 95% confidence intervals around the mean. All scale bars represent 20 μM.

The online version of this article includes the following source data and figure supplement(s) for figure 6:

*Figure 6 continued on next page*

*Figure 6 continued*

**Source data 1.** This spreadsheet contains the raw data for the HRP defects plotted in *Figure 6*.

**Figure supplement 1.** Nedd4 and Smurf are expressed in the nerve cord at developmental stages when midline crossing occurs.

**Figure supplement 2.** Smurf does not appear to be required for midline crossing in vivo.

**Figure supplement 2—source data 1.** This is the raw data used to produce the localization graphs in *Figure 6*, *Figure 6—figure supplement 2*.

**Figure supplement 3.** Su(dx) does not appear to be required for midline crossing in vivo.

**Figure supplement 3—source data 1.** This is the raw data used to produce the localization graphs in *Figure 6*, *Figure 6—figure supplement 3*.

fra single mutants) and most of the remaining commissures are thin (*Figure 6A and B*). These defects are also observed in the anterior commissure, a phenotype that is never observed in *fra* mutants alone. The crossing defects of the eagle neurons are also significantly increased and include defects in the projection of Eg neurons through the anterior commissure (*Figure 6C and D*). These observations indicate that *Nedd4* acts in parallel to *fra* to promote commissural axon growth across the midline.

To further support these observations, we also examined the consequences of removing *nedd4* In the FraΔC background. Expression of FraΔC using the egGal4 driver induces EW crossing defects in 30–50% of abdominal segments but does not affect EG neuron crossing. While Nedd4 heterozygotes show no alterations in commissural axon guidance defects in the FraΔC background, Nedd4 homozygous mutants display significantly increased EW non-crossing phenotypes. We were able to take advantage of this genetic interaction to test for a neuron-autonomous function of Nedd4 by performing rescue experiments. We find that expressing a UAS Nedd4 transgene in embryos homozygous mutant for *nedd4* in the FraΔC background leads to a significant reduction in the non-crossing defects (*Figure 6E and F*). Taken together, these results demonstrate an important role for Nedd4 in promoting midline crossing and suggest that it functions cell autonomously in neurons.

## Smurf and Su(dx) are dispensable for midline crossing

There are three Nedd4 family E3 ubiquitin ligases in *Drosophila:* Nedd4, Smurf, and Su(Dx). To investigate whether Smurf and Su(dx) could also play a role in midline crossing, we first examined their expression patterns by searching the embryonic mRNA expression data base of the Berkeley *Drosophila* Genome Project (BDGP). This database includes images of mRNA in situ patterns for many fly genes. Data reported for Su(Dx) show that it is expressed broadly or ubiquitously throughout all stages of embryogenesis including in the nerve cord of stages 14 through 16 embryos. No data is publicly available for Smurf, so we generated our own in situ probes to visualize the Smurf embryonic mRNA expression pattern and found that like Su(Dx), Smurf is broadly expressed during embryogenesis, including in the developing nerve cord (*Figure 6—figure supplement 1*). Thus, although Su(Dx) and Smurf do not show striking nerve cord enrichment like Nedd4, their expression patterns are consistent with potential roles in commissural axon guidance.

To investigate roles for Smurf and Su(Dx) in midline axon guidance, we examined the nerve cords of embryos lacking functional copies of each gene. We used the Smurf 15 C loss-of-function allele, generated by insertion of a hobo element which disrupts the coding sequence of its HECT domain (*Podos et al., 2001*), and the deficiency line Df(2 L)Exel7008 which contains the smallest available chromosomal deletion covering the Su(dx) locus. Similar to *nedd4* mutants, zygotic loss of either Smurf or Su(dx) function produced no visible midline crossing defects in nerve cords stained with the pan-neuronal marker HRP (*Figure 6*, *Figure 6—figure supplement 2A, B* and *Figure 6—figure supplement 3*). Due to potential effects of maternal deposition and possible redundancy, we next investigated whether loss of Smurf or Su(dx) could induce midline crossing defects in the *fra* mutant and FraΔC sensitized genetic backgrounds as described above for our investigation of *nedd4*. In marked contrast to our findings with *nedd4*, removing both zygotic copies of either *smurf* or *su(dx)* does not increase the occurrence of EW non-crossing in either *fra* mutant embryos or in embryos expressing FraΔC using egGal4 (*Figure 6*, *Figure 6—figure supplement 2C-F* and *Figure 6—figure supplement 3*). Based on these findings, it is unlikely that either ligase plays a major role in midline crossing; however, we cannot rule out the possibility that either Smurf or Su(Dx) acts redundantly with Nedd4. Nevertheless, our genetic experiments are most consistent with Nedd4 playing the prominent role in regulating midline crossing.

## Nedd4 interacts with the slit/Robo pathway both biochemically and genetically

Our observations that Comm facilitates Robo1 ubiquitination and lysosomal degradation through its cytoplasmic PY motifs, together with our genetic data showing that *Nedd4* promotes commissural axon guidance across the midline, suggest that Comm and Nedd4 function together to inhibit Robo1 repulsion. If indeed, Comm serves as a substrate adapter to allow Nedd4 to target Robo1 for ubiquitination and subsequent degradation, we reasoned that the three proteins should form a complex. Furthermore, we predicted that in the absence of Comm or when the PY motifs of Comm are mutated, Nedd4 should exhibit reduced association with the Robo1 receptor. To test these predictions, we performed a series of coimmunoprecipitation experiments in S2R + cells. When Robo1 and Nedd4 are co-transfected in the absence of Comm, we detect very little if any association between Nedd4 and Robo1 (*Figure 7A*). Strikingly, when all three proteins are expressed, Nedd4 readily co-precipitates with Robo1, even though the presence of Comm greatly reduces Robo1 protein levels (*Figure 7A*). As predicted, mutating the PY motifs leads to a significant decrease in the amount of Nedd4 that is incorporated into the complex, even though the level of Robo1 protein expression increases when Comm PY motifs are mutated (*Figure 7B*). Both observations are consistent with the model that Comm acts as a substrate adapter between Nedd4 and Robo1 (*Figure 7C*).

To determine whether Nedd4 can function together with Comm in vivo to downregulate the expression of the Robo1 receptor, we first investigated whether Robo1, Comm, and Nedd4 could form a three-member complex like that we observed in cells, and found that in lysate of embryos co-expressing 10 X UAS WT Comm and 10 X UAS Nedd4, we were able to immunoprecipitate a complex consisting of these two proteins and endogenous Robo1 (*Figure 7—figure supplement 1*). To test whether Nedd4 enhances comm-induced Robo1 degradation, we measured the expression level of endogenous Robo1 protein biochemically in wild-type embryos, embryos expressing 10 X UAS Comm, or embryos expressing both 10 X UAS Comm and 10 X UAS Nedd4. We collected protein lysates from these three groups of embryos that were aged between 13 and 18 hours postfertilization, since this is developmental window when axons in the CNS are actively growing toward their targets. As predicted from our in vitro measurements of the effect of Comm on Robo1 expression levels, pan-neural expression of Comm leads to a striking reduction in Robo1 protein levels. Importantly this effect is significantly enhanced in embryos co-expressing Comm and Nedd4 (*Figure 7D and E*), which further supports the idea the Nedd4 facilitates Comm-dependent Robo1 degradation. Interestingly, and consistent with our in vitro biochemistry, co-expression of Nedd4 and Comm also leads to reduced levels of Comm protein relative to embryos expressing Comm alone, suggesting that Comm, like Robo1 is likely degraded due to ubiquitination from Nedd4 (*Figure 7E*, see discussion).

While these observations strongly support the idea that Nedd4 can work with Comm to downregulate Robo1 protein levels in vivo, they do not allow us to make conclusions about where Robo1 levels are decreasing. To address this question, we monitored endogenous Robo1 levels in these embryos by immunofluorescence to determine if co-expression of Comm and Nedd4 leads to reduction in the axonal expression or Robo1. Consistent with the data presented in earlier (*Figure 3* with accompanying supplement, and 4), ectopic expression of 5 X UAS Comm results in a striking decrease in axonal Robo1 expression. Importantly, axonal Robo1 levels are further reduced when Nedd4 is co-expressed with Comm (*Figure 7F–G*). Taken together, these findings support a model in which Comm recruits Nedd4 via its PY motifs, bringing it into proximity with Robo1, facilitating its ubiquitination, localization to late endosomes/lysosomes, and ultimately its lysosomal degradation (*Figure 8*).

Because Nedd4 enhances reduction of Robo1 levels via Comm, we finally sought to investigate whether Nedd4 increases Comm-induced ectopic crossing, which is a morphological output of reduced Robo1 signaling. To do so, we examined and quantified ectopic crossing phenotypes in WT embryos, embryos pan-neuronally expressing 5 X UAS WT Comm, and embryos co-expressing 5 X UAS WT Comm and 10 X UAS Nedd4 (*Figure 7—figure supplement 2*). The factors we quantified were nerve cord width (*Figure 7—figure supplement 2B*) and the collapse of segments (*Figure 7—figure supplement 2D*) within the nerve cord. To compare nerve cord widths across genotypes, we measured the width of the eight posterior-most body segments and calculated the average. We then binned nerve cord widths into the following three phenotypic categories:≤25 μm, 25<x < 30, and ≥30. Distribution of phenotypes among CTRL embryos is significantly different from that within the population of embryos expressing either 5 X Comm alone, or 5 X Comm +10 X Nedd4 (*Figure 7—figure*

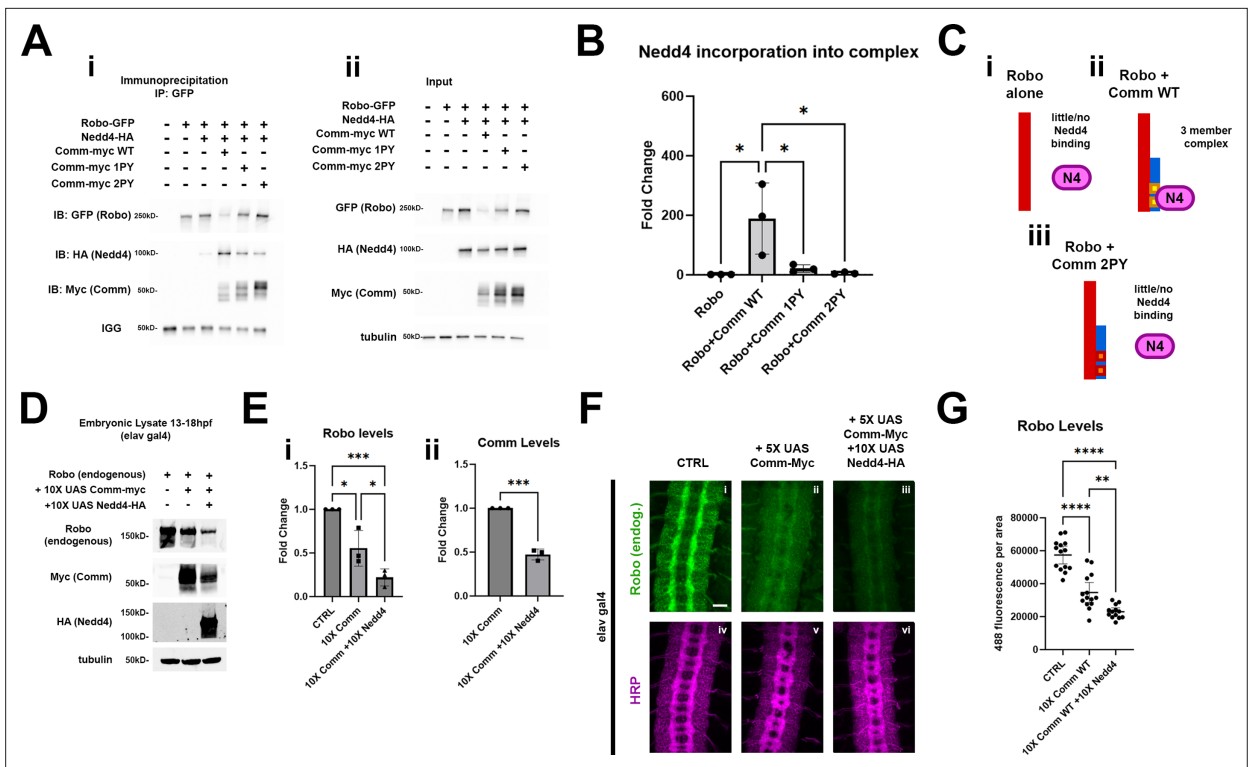

**Figure 7.** Nedd4 interacts genetically and biochemically with slit/Robo1 pathway. (**A–C**) Nedd4 is incorporated into a three-member complex with Robo1 and Comm. (**A**) Immunoprecipitation (**i**) and input (**ii**) of Robo1-GFP in S2*R* + cells co-transfected with Nedd4-HA and WT, 1PY, or 2PY Comm-myc variants. (**B**) Quantification of fold change in Nedd4 association with Robo1, relative to the Robo1 + Nedd4 condition, from the co-immunoprecipitation in (**A**). This was measured by quantifying immunoprecipitated Nedd4 via densitometry and normalizing to immunoprecipitated Robo1 and input Nedd4. Groups were compared using ANOVA with Tukey's post-hoc test. (* p<0.05) Error bars represent mean +/-one standard deviation. (**C**) Schematic illustrating how Nedd4 relies on Comm PY motifs for recruitment into a ternary complex with Robo1 and Nedd4. (**D–G**) Exogenous Nedd4 enhances degradation of Robo1 in the presence of WT Comm in vivo. (**D**) Western blot of Lysates of 13-18hpf embryos expressing WT Comm-myc with or without Nedd4-HA under the elavGal4 driver. (**E**) Quantification of Robo1 (**i**) and Comm (**ii**) levels from the blot in (**D**). Protein levels were measured via densitometry and normalized to tubulin. Protein levels were compared across conditions using ANOVA with Tukey's post-hoc test. (* p<0.05, *** p<0.001, **** p<0.0001) Error bars represent mean +/-one standard deviation. (**F**) Nerve cords of stage 15–16 embryos expressing WT Comm-myc with or without Nedd4-HA under the elavGal4 driver. Embryos are stained with the pan-neuronal marker HRP (magenta) and an antibody against endogenous Robo1 (green). Scale bar represents 20 μM. (**G**) Quantification of Robo1 levels from the conditions in (**F**). Robo1 levels were measured by creating a mask of the axonal scaffold, measuring 488 fluorescence within the scaffold, and dividing by scaffold area. Robo1 levels were compared across conditions using ANOVA with Tukey's post-hoc test. (** p<0.01, **** p<0.0001) Error bars represent 95% confidence intervals around the mean.

The online version of this article includes the following source data and figure supplement(s) for figure 7:

**Source data 1.** This zip file contains the raw western blots and requested annotated files for *Figure 7*.

**Source data 2.** This zip file contains the raw western blots and requested annotated files for *Figure 7*.

**Source data 3.** This zip file contains the raw western blots and requested annotated files for *Figure 7*.

**Source data 4.** This zip file contains the raw western blots and requested annotated files for *Figure 7*.

**Figure supplement 1.** Robo, Comm, and Nedd4 can form a three-member complex in fly embryonic lysate.

**Figure supplement 1—source data 1.** This zip file contains the raw western blots and requested annotated files for *Figure 7*.

**Figure supplement 1—source data 2.** This zip file contains the raw western blots and requested annotated files for *Figure 7*, *Figure 7—figure supplement 1*.

**Figure supplement 2.** Nedd4 enhances ectopic crossing phenotype induced by Comm overexpression.

**Figure supplement 2—source data 1.** This is a spreadsheet with the raw data for the quantification of nerve cord width in *Figure 7*, *Figure 7—figure supplement 2*.

**Figure supplement 2—source data 2.** This is a spreadsheet with the raw data for the quantification of nerve cord collapse in *Figure 7*, *Figure 7—figure supplement 2*.

**Figure supplement 2—source data 3.** This is a spreadsheet with the raw data for the quantification of nerve cord collapse measurements in *Figure 7*, *Figure 7—figure supplement 2*.

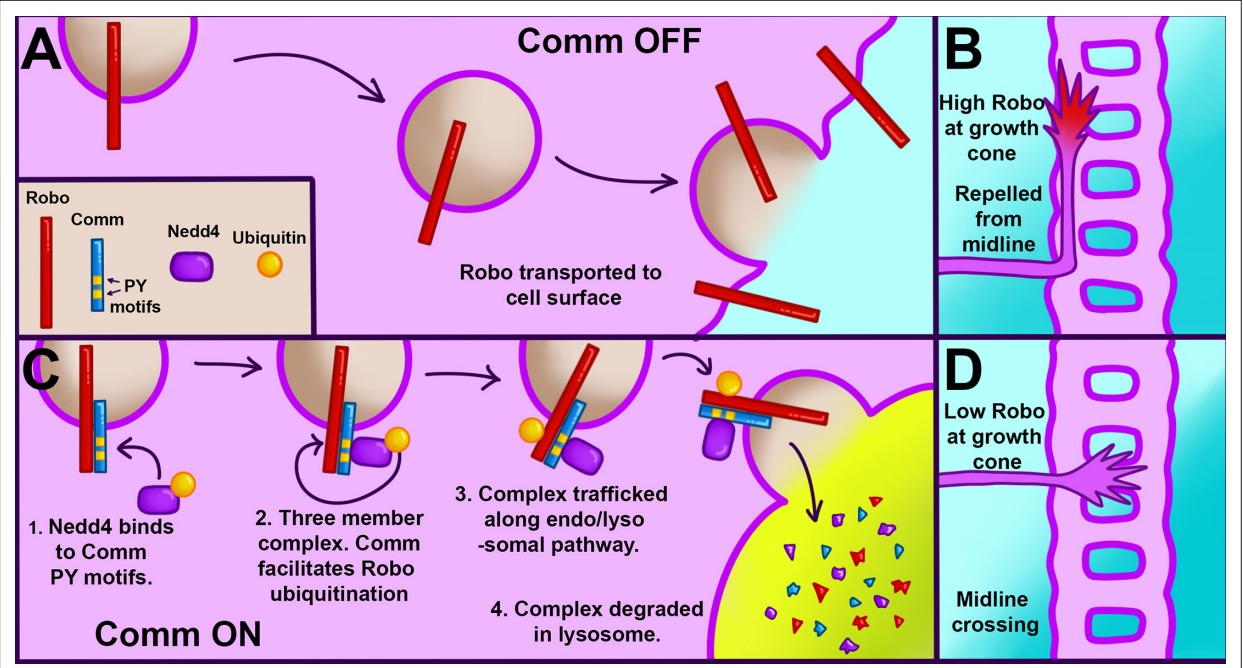

**Figure 8.** A schematic illustrating our proposed model for Robo1 downregulation in pre-crossing commissural axons. (**A–B**) Localization and repulsive activity of Robo1 in neurons when Comm is absent. (**A**) View of Robo1 trafficking down axons to the growth cone surface. In neurons not expressing Comm (such as ipsilateral or post-crossing neurons), Robo1 travels unimpeded to the growth cone surface. (**B**). At the growth cone surface, Robo1 induces a repulsive response to slit, preventing midline crossing. (**C–D**) Robo1 localization and repulsive activity in the presence of Comm. (**C**) Comm facilitates Robo1 ubiquitination by recruiting Nedd4 via its PY motifs. Robo1 ubiquitination drives the entire Robo1/Nedd4/Comm complex to the lysosome. (**D**) Due to lysosomal degradation, little Robo1 reaches the growth cone membrane. This lack of Robo1 renders axons insensitive to repulsion via slit, enabling them to cross the midline.

supplement 2B). While 95% of CTRL embryos (n=20) have a nerve cord width of over 30 µM, all nerve cords of embryos expressing either 5 X Comm alone (n=17), or 5 X Comm +10 X Nedd4 (n=17) were less than 30 µM. In addition, distribution of phenotypes among embryos expressing 10 X comm alone was significantly different from that of embryos co expressing 5 X Comm and 10 X Nedd4. All nerve cords of embryos expressing 5 X Comm alone were between 25 µm and 30 µm wide (*Figure 7— figure supplement 2B*), while the nerve cords of embryos co-expressing 5 X Comm and 10 X Nedd4 could be divided into two different phenotypic classes (*Figure 7—figure supplement 2A*). In these embryos, 64.7% (11 of 17 total) nerve cords resembled those of embryos expressing 5 X Comm alone and had widths between 25 and 30 µM. A second class of embryos (6 of 17 total) exhibited a more severe "extra thin" phenotype with widths less than 25 µM (*Figure 7—figure supplement 2B*). These findings demonstrate that Nedd4 can enhance reduction of nerve cord width induced by overexpression of Comm.

In addition to nerve cord width, we examined another morphological defect associated with ectopic crossing, the collapse of nerve cord segments defined by loss of negative space between their 'ladder rungs' (normal and collapsed segments illustrated in *Figure 7—figure supplement 2C*, collapsed segments indicated with yellow arrowheads in *Figure 7—figure supplement 2A* micrographs). For our analysis, we binned embryos into three different phenotypic classes: no collapse, partial collapse (at least one segment collapsed), and total collapse (100% of segments collapsed). We found that the distribution of phenotypes within WT embryos differed significantly from that within 5 X Comm-expressing embryos, as well as that of embryos co-expressing 5 X Comm (p<0.0001) and 10 X Nedd4 (p<0.0001 *Figure 7—figure supplement 2D*). Nerve cords of all WT embryos (n=20) showed no collapse (*Figure 7—figure supplement 2A*), while only 41.2% (7 of 17 embryos) of embryos expressing 10 X Comm alone and 35.3% (6 of 17 total) of embryos expressing 5 X Comm +10 X Nedd4 exhibited this phenotype, respectively (*Figure 7—figure supplement 2D*). Distribution of phenotypes between embryos expressing 5 X Comm alone and embryos co-expressing 5 X Comm and 10 X Nedd4 also differed significantly (p<0.05, *Figure 7—figure supplement 2D*). In embryos

expressing 10 X comm, all nerve cords exhibiting collapse were binned into the 'partial collapse' class (11 of 17 total). Meanwhile, in embryos co-expressing 5 X Comm and 10 X Nedd4, 29.4% nerve cords (5 of 17 total) exhibited partial collapse phenotypes and 35.3% (6 of 17 total) of nerve cords exhibited complete collapse (*Figure 7—figure supplement 2A and D*). The presence of this 'complete collapse' phenotypic class demonstrates that addition of Nedd4 enhances the severity of morphological defects induced by Comm overexpression. Taken together with our nerve cord width data, these findings show that Nedd4 enhances the Comm gain-of function phenotype defined by ectopic crossing, likely by enhancing downregulation of Robo1 levels.

## Discussion

The data presented here support a model that is a hybrid of the two developed previously (*Keleman et al., 2005*; *Myat et al., 2002*), in which Comm requires interaction with Nedd4 to downregulate Robo1, and in which Robo1, not Comm, is the primary substrate for Nedd4 ubiquitination. Through a combination of in vivo and in vitro experiments, we demonstrate that Comm requires intact PY motifs to localize effectively to late endosomes/lysosomes, route Robo1 away from the cell surface, facilitate Robo1 ubiquitination and degradation, and enable midline crossing. We also find that Comm lacking one or both of its PY motifs is more stable than its wild-type counterpart. Using in vitro biochemical assays, we show that Nedd4 is recruited into a ternary complex with Comm and Robo1 in a manner dependent on Comm PY motifs. In addition, in-vivo loss and gain-of-function experiments indicate a specific and cell-autonomous role for Nedd4, but not other Nedd4 family members Su(dx) and Smurf, in midline crossing. Finally, we show that overexpression of Nedd4 enhances downregulation of Robo1 levels by Comm in vivo, as well as turnover of the Comm protein itself. Taken together, these findings support a model of Robo1 downregulation in which Comm acts as an adaptor protein between Robo1 and Nedd4, recruiting Nedd4 into a three-member complex with itself and Robo1 via its PY motifs. Once the two proteins are close to one another, Nedd4 can ubiquitinate Robo1, leading to lysosomal trafficking and degradation of the entire complex and ultimately, inhibition of premature Robo1-mediated repulsion at the growth cone (*Figure 8*).

### New insights into the role of PY motifs in comm function

In accordance with previous studies (*Keleman et al., 2002*; *Myat et al., 2002*), we find that PY motifs are necessary for Comm to promote ectopic midline crossing in ipsilateral neuron populations and to promote Comm localization to intracellular puncta in vitro. One study had reported that mutation of the LPSY motif alone completely disrupts both Comm's ability to promote ectopic crossing and its localization to presumptive endosomes in cultured cells (*Keleman et al., 2002*). In contrast, by directly comparing Comm transgenes inserted in the same genetic locus, we were able to dissect the relative roles of these motifs and show that the PY motifs act in an additive manner, as disruption of the LPSY motif alone and both PY motifs disrupted Comm's ability to promote midline crossing in a dose-dependent manner. Our experiments in vitro further support an additive role for the PY motifs in controlling Comm and Robo1 localization and suggest distinct activities of the two motifs. Specifically, we observe that while Comm 2PY is unable to efficiently localize itself or Robo1 to intracellular puncta in COS-7 cells, Comm lacking only the LPSY motif itself localizes normally, but it is compromised in its ability to route Robo1 to puncta along with it. This suggests the interesting possibility that there may be instances when Comm and Robo1 can separate during intracellular trafficking. We further dissected the structural requirements for Comm localization in vivo, where we found that disruption of Comm PY motifs causes 'leakage' of Comm from cell bodies into axons in a dose-dependent manner. We also observe that Robo1 colocalizes equally with wild-type and PY mutant Comm variants, and mimics the localization of the Comm variant with which it is co-expressed, suggesting that comm PY motifs play a role in keeping Robo1 from entering the axon. These observations suggest that PY motifs are required to shunt Robo away from the growth cone and this possibility is further supported by our finding that Comm lacking functional PY motifs is unable to reduce the expression of Robo1 on the surface of commissural axons in vivo. Finally, we demonstrate that mutation of the LPSY or both PY motifs compromises colocalization of Comm and Lamp1 + puncta, showing that PY motifs are required for trafficking to late endosomes and lysosomes in vivo.

In addition to localization, our work offers important new insights into how Comm downregulates Robo1 protein levels. Both in vivo and in vitro, we find that disrupting Comm PY motifs inhibits its ability to promote Robo1 degradation. Biochemical assays demonstrate that Comm requires PY motifs to recruit Nedd4 into a complex with itself and Robo1 and promote Robo ubiquitination. As ubiquitinated Robo1 is stabilized by chloroquine, our results show that Comm facilitates the degradation of this pool of Robo1 via a lysosomal pathway. Taken together, these observations implicate PY motifs as major players in all known aspects of Comm function: localization to late endosomes, ability to regulate localization and protein levels of its cargo, and promotion of midline crossing.

## Robo1 is a substrate for Nedd4 ubiquitination

Comm has been shown to be ubiquitinated in the presence of Nedd4 *Myat et al., 2002*; however, the importance of ubiquitination for Comm function has been cast into doubt by the findings that an un-ubiquitinatable form of Comm (Comm KR) is able to localize correctly and traffic Robo1 to intracellular puncta in COS-7 cells, and promote midline crossing in vivo as effectively as WT Comm (*Keleman et al., 2005*). Despite these observations and studies implicating ubiquitination as a negative regulator of Robo1 repulsion (*Yuasa-Kawada et al., 2009*), the possibility that Comm facilitates ubiquitination of its cargo Robo1 has not yet been explored. We found that in the presence of Comm, Robo1 ubiquitination is significantly increased, and that mutation of Comm PY motifs prevents this increase. This, combined with the observation that the addition of exogenous Nedd4 leads to a further reduction of Robo1 protein levels by Comm and an enhancement of ectopic crossing phenotypes, supports a model in which Comm downregulates Robo1 by recruiting Nedd4 through its PY motifs to trigger Robo1 ubiquitination and subsequent lysosomal degradation. In our biochemical assays, mutation of Comm PY motifs not only impedes downregulation of Robo1 protein levels but also increases the stability of Nedd4 (*Figure 7Aii*) and Comm, suggesting that these three proteins are trafficked to the lysosome and degraded together as a complex.

## Nedd4 is the primary member of the Nedd4 family involved in midline crossing

Prior studies had generated contradictory findings concerning the role for Nedd4 in midline axon guidance. Here, we provide conclusive evidence of a role for Nedd4 in promoting midline crossing by showing that loss of *nedd4* increases crossing defects in the FraΔC or fra-/- sensitized genetic backgrounds. Further, we show that in the FraΔC background, re-expression of transgenic Nedd4 in a the Eg subset of commissural neurons rescues the midline crossing defects, indicating a likely cell autonomous role for Nedd4 in this process. In contrast, zygotic mutation of either Su(dx) or Smurf did not result in midline axon guidance defects in wild-type embryos or in either of these sensitized genetic backgrounds. While we were unable to show evidence of a role for Su(dx) and Smurf through genetic assays in sensitized backgrounds, we still have insufficient evidence to definitively rule out a role for them. Many of the factors prompting us to use sensitized backgrounds, especially maternal expression and redundancy among Nedd4 family members, present us with significant challenges in detecting loss of function phenotypes. In addition, Nedd4 family E3 ligases regulate several additional pathways involved in critical early-developmental patterning events, including germ layer specification, establishment of major body axes, and fate determination of the neuroectoderm. At least two different Nedd4-family ligases are known to regulate each of the following pathways: Notch, TGF-β, and Wnt (for review see *Ingham et al., 2004*). The fact that none of these pathways are regulated by a sole Nedd4 family E3 ligase suggests that Nedd4 may not be the only Nedd4 family member regulating Robo1 and Comm. Unfortunately, the conditions for definitively testing their involvement, such as mutating multiple Nedd4 family members simultaneously or abolishing their maternal expression has not yet been possible.

## Comm is likely to regulate Robo1 repulsion through multiple mechanisms

Our data clearly supports an important role for Nedd4 in contributing to Comm's ability to inhibit Robo1 signaling and promote midline crossing; however, several lines of evidence support alternative roles for Comm in preventing midline repulsion. First, a previous study investigated the consequences of generating an endogenously expressed Robo1 receptor that cannot be inhibited by Comm through

homologous recombination. Specifically, the coding sequence of Robo1 was replaced by a mutant variant that could not bind to or be downregulated by Comm (termed Robo1-SD for sorting defective; *Gilestro, 2008*). If Comm's sole function were to inhibit the surface expression of Robo1 then the prediction would be that these Robo1-SD embryos would phenocopy *comm* mutants; however, quite surprisingly, these embryos appear to be completely normal (*Gilestro, 2008*). This finding suggests either Comm inhibits the function of another key component of the Slit-Robo1 repulsive pathway, potentially through inhibiting its expression, or alternatively, Comm may inhibit Robo1 function through a second mechanism that is independent of ubiquitin-mediated protein degradation. Since Robo1-SD does not bind to Comm, this alternative mechanism would not depend on binding to Robo1. Our finding that high-level over-expression of the Comm2PY mutant variant that cannot inhibit Robo1 surface expression can still inhibit midline repulsion is consistent with mechanism of regulation that is independent of protein degradation of Robo1, or another Comm substrate. One interesting potential alternative role for Comm would be to prevent Robo1 endocytosis, since receptor endocytosis is required for repulsive signaling (*Chance and Bashaw, 2015*; *Sullivan and Bashaw, 2023*). Indeed, at the *Drosophila* neuromuscular junction (NMJ), Comm has been proposed to regulate endocytosis from the plasma membrane to promote synapse formation (*Ing et al., 2007*; *Wolf et al., 1998*).

## The role for Nedd4 is evolutionarily conserved

Although we still do not know whether Smurf or Su(dx) play roles in midline crossing, our observation that Nedd4 appears to be the main ligase involved in this process agrees with recent findings in mice and provides additional evidence of a conserved mechanism of Robo1 downregulation between vertebrates and invertebrates. While Comm appears to be only present in insects, several vertebrate proteins have been identified that may play analogous roles in mammalian species (*Gorla et al., 2019*; *Justice et al., 2017*). Among these proteins that may function analogously to Comm, the Nedd4 family interacting proteins Ndfip1 and Ndfip2 share sequence similarity to the region of Comm containing PY motifs and appear to act as its functional analog in mice. Like Comm, loss of mammalian Ndfip proteins impedes effective midline crossing and results in the upregulation of Robo1 levels in pre-crossing commissural axons in the spinal cord. In addition, Ndfip proteins negatively regulate Robo1 expression in a manner nearly identical to that of Comm: by facilitating Robo1 ubiquitination, diversion away from the cell surface, trafficking to late endosomes, and ultimately, degradation (*Gorla et al., 2019*).

Since these functions of Ndfip proteins rely on functional PY motifs, efforts have been made to determine which of the nine mammalian Nedd4 family E3 ligases they may recruit to regulate Robo1 localization and protein levels. While the seven E3 ligases most related to *Drosophila* Nedd4 family members can all bind Ndfip proteins in vitro, only the Nedd4 homologs Nedd4-1 and Nedd4-2, and the Su(dx) homolog WWP2 can enhance Robo1 ubiquitination and degradation. Unlike the other Nedd4 family ligases tested, Smurf homologs Smurf1 and Smurf2 can bind to Ndfip proteins in the absence of functional PY motifs suggesting that they bind to additional structural elements of Ndfip proteins and may be involved in different biological processes. In addition, single and double conditional knockouts of Nedd4-1 and Nedd4-2 display impaired axon guidance across the floor plate of the spinal cord. Taken together, these findings demonstrate an important role for mammalian Nedd4 homologs in commissural axon guidance in the spinal cord (*Gorla et al., 2022*). The finding that some Nedd4 family members do not facilitate Robo1 ubiquitination and degradation despite binding effectively to Ndfip proteins suggests that additional factors are required to confer substrate specificity to E3 ligases. These additional factors appear to be highly cell-type specific, as the same cargo can be ubiquitinated by different E3 ligases in different cellular contexts. For example, Ndfip-mediated ubiquitylation of the divalent metal ion transporter (DMT1) is catalyzed by Nedd4-2 in neuronal cells (*Howitt et al., 2009*), but is catalyzed by WWP2 in other non-neuronal cell types (*Foot et al., 2008*; *Gorla et al., 2022*). What factors may determine E3 ligase substrate specificity in the developing mammalian and insect CNS remains an open question.

Despite the lack of comm homologs outside of insects, the work shown in this paper and the research conducted on Ndfip proteins in the mouse spinal cord point to a common mechanism of Robo1 regulation across species. In both vertebrates and invertebrates, Nedd4 substrate adaptor proteins serve as a vital means of downregulating growth cone Robo1 levels by facilitating Robo1 ubiquitination, trafficking to lysosomes, and ultimately, degradation. Discovering additional binding

partners for both Comm and Ndfip proteins, especially those shared between both, will be necessary for us to better understand other critical players in this common mechanism. Such binding partners may include proteins involved in determining E3 substrate specificity and in trafficking Nedd4 adaptors and their cargoes to the proper destinations. As it is highly likely that Comm regulates other proteins besides Robo1, and Ndfip has several known cargoes in different cell contexts (*Foot et al., 2008*; *Kang et al., 2015*; *Mund and Pelham, 2010*; *Trimpert et al., 2017*), additional binding partners may also include alternative cargoes of Nedd4 adaptors involved in axon guidance. Such an investigation may offer important insights into how this fundamental Nedd4 substrate adaptor-mediated mechanism to tune the neuronal surface proteome may contribute to development and function of the nervous system.

## Materials and methods
### Contact for reagent and resource sharing
Further information and requests for resources and reagents should be directed to the Lead Contact, Greg J. Bashaw (gbashaw@pennmedicine.upenn.edu).

### Experimental model and subject details
#### Genetic stocks
The following *Drosophila* strains were used: *w1118*, *apGal4*, *egGal4*, *elavGal4*, *UAS-CD8GFP II*, *UAS-TauMycGFP III*, *UAS-fraΔc-HA*, *10UAS-Robo1-HA*, *fra3*, *nedd4T119FS*, *Mi{PT-GFSTF.0}Nedd4[MI07766-GFSTF.0]*, *Df(2 R)ED3385*, and *Df(2 R)Excel7008*. The fly strain *smurf15C* was a kind gift from the Pi Lab at Chang Gung University. The following transgenic stocks were generated: *10UAS-Comm-myc WT*, *10UAS-Comm-myc 1PY*, *10UAS-Comm-myc 2PY*, *5UAS-Comm-myc WT*, *5UAS-Comm-myc 1PY*, *5UAS-Comm-myc 2PY*, and *10UAS-Nedd4-HA*. Transgenic flies were generated by Bestgene Inc (Chino Hills, CA) using PhiC31- directed site-specific integration into landing sites at cytological position 86F8 (For *UAS-Comm* constructs) or 51 C (for *UAS-Nedd4* constructs). All crosses were carried out at 25 °C.

#### Cell culture
*Drosophila* S2R + cells were maintained at 25 °C in Schneider's media (Life Technologies, #21720024) supplemented with 10% (vol/vol) FBS and a mixture of 1% Penicillin and Streptomycin. COS-7 cells were maintained at 37 °C, 5% $CO_2$ in Dulbecco's Modified Eagle Medium (Corning, 10–017 CM) supplemented with 10% (vol/vol) FBS and a mixture of 1% Penicillin and Streptomycin.

### Method details
#### Molecular biology
To make the P10UAST-Comm-myc WT, the WT coding sequence of Comm was PCR amplified and cloned into the *p10UAST-myc* vector described above by swapping out the *hspc300* insert using *Kpn1/Asc1* sites and the following primers:

> Fwd with Kpn1: GCGCGGTACCATGATTAGCACCACGGATTAT
> Rev with Asc1: AAAGGCGCGCCCGCGGCAACAACAACGA
> To make the P10UAST-Comm-myc 1PY construct, the LPSY motif of Comm was mutated to AASY via SOE-ing PCR, using the WT Comm coding sequence from P10UAST-Comm-myc WT as a template. The following primers were used to flank the coding sequence and introduce point mutations into the LPSY motif.
> Fwd with Kpn1: GCGCGGTACCATGATTAGCACCACGGATTAT
> Rev with Asc1: AAAGGCGCGCCCGCGGCAACAACAACGA
> LPSY->AASY Fwd: AATCGCCACCGGAGCGGCCAGCTACGATGAGGCACTGCAT
> LPSY->AASY Rev: ATGCAGTGCCTCATCGTAGCTGGCCGCTCCGGTGGCGATT

The resulting Comm 1PY coding sequence was then subcloned back into a n-terminal myc-tagged P10UAST backbone using Kpn1 and Asc1 restriction sites.

To generate the P10UAST-Comm-myc 2PY, the coding sequence from P10UAST Comm-myc 1PY was subcloned into the smaller pbluescript backbone. Point mutations were introduced into the PPCY motif using Quikchange II site-directed mutagenesis kit (Agilent, #200523) using the following primers.

CAAATCGAATCGGCGGCCTGCTACACAATCGCCACCGGAT
ATCCGGTGGCGATTGTGTAGCAGGCCGCCGATTCGATTTG

The resulting Comm 1PY coding sequence was then subcloned back into a n-terminal myc-tagged P10UAST c-term myc backbone using Kpn1 and Asc1 restriction sites.

To generate p5UAST Comm-myc constructs, 5 UAS sites were excised from the P10UAS Comm-myc constructs using the Sph1 restriction enzyme.

To generate myc-Comm constructs for transfection in mammalian cells, the coding sequences of WT, 1PY, and 2PY Comm from the P10UAST Comm-myc constructs were subcloned into the pKMyc backbone (Addgene #19400) using the Xba1 and Nhe restriction sites and the following primers:

Fwd with Xba1: CCGGTCTAGAATGATTAGCACCACGGATTA
Rev with Nhe1: ATAGCTAGCTCACGCGGCAACAACAA

To make the P10UAST-FLAG-Ub construct, the FLAG-Ubiquitin coding sequence from a mammalian-expressing Flag-Ub plasmid (cited in *Gorla et al., 2019*, originally a kind gift from the Dr. Hideaki Fujitha lab) was subcloned into the P10UAST backbone using Xho1 and Xba1 restriction sites. The following primers were used.

Fwd with Xho1: AAACTCGAGATGGACTACAAAGACCATGA
Rev with Xba1: AAATCTAGATTAACCACCACGAAGTCTCA

The P10UAST-Nedd4-HA construct was made by PCR amplifying Nedd4 cdna from the pOT2 DGRC Gold collection (clone SD04682, DGRC Stock 5253) using the following primers.

Fwd with Not1: ATATGCGGCCGCATGTCGGCACGTTCCAGC
Rev with Kpn1: CCCGGTACCATCAACTCCAGCAAATCCTTGGC

The resulting insert was then subcloned into the P10UAST-HA backbone using Not1 and Kpn1 restriction sites.

## Western blot

S2R + cells were transiently transfected with Effectene transfection reagent (QIAGEN, Valencia CA, #301425) and induced 24 hr later with 0.5 mM copper sulfate. 24 hr after induction, cells were lysed in TBS-V (150 mM NaCl, 10 mM Tris pH-8, 1 mM ortho-vanadate) supplemented with 0.5% Surfact-AMPS NP40 (Thermo, Waltham MA, #85124) and 1 x Complete Protease Inhibitor (Roche, #11697498001) and incubated with agitation for 20 min at 4 °C. Soluble proteins were recovered by centrifugation at 15,000 rpm for 15 min at 4 °C.

To collect lysates from *Drosophila* embryos, approximately 100 µl of embryos at the desired developmental stages were treated with 50% bleach (Essendant) for 2 min and washed three times with embryo wash buffer (120 mM NaCl supplemented with 0.1% (v/v) Triton X-100, Sigma). Embryos were then rinsed with ice cold TBSV buffer (150 mM NaCl, 10 mM Tris pH 8.0, 2 mM Sodium orthovanadate), and lysed with manual homogenization using plastic pestles in 300 µl lysis buffer (TBSV supplemented with 1% Surfact-AMPS NP40 and 1 x complete protease inhibitors). Homogenized samples were then incubated for 30 min at 4 °C with agitation and recovered by centrifugation at 15,000 rpm for 15 min 4 °C.

Proteins were resolved by SDS-PAGE and transferred to nitrocellulose membrane (Amersham, #10600032). Membranes were blocked with 5% dry milk and 0.1% Tween 20 in PBS for 1 hr at room temperature and incubated with primary antibodies in 2% dry milk/0.1% Tween 20 in PBS overnight at 4 °C. Following three washes with PBS/0.1% Tween 20, membranes were incubated with the appropriate HRP-conjugated secondary antibody at room temperature for 1 hr. Signals were detected using Clarity ECL (Bio-Rad, #1705061) or Super Signal West Femto Maximum Sensitivity Substrate (Thermo Fisher, 34094) according to manufacturer's instructions.

## Immunoprecipitation

S2R + cells were transiently transfected with Effectene transfection reagent (QIAGEN, Valencia CA, #301425) and induced 24 hr later with 0.5 mM copper sulfate. 24 hr after induction, cells were lysed in TBS-V (150 mM NaCl, 10 mM Tris pH-8, 1 mM ortho-vanadate) supplemented with

0.5% Surfact-AMPS NP40 (Thermo, Waltham MA, #85124) and 1 x Complete Protease Inhibitor (Roche, #11697498001) for 20 min at 4 °C. Soluble proteins were recovered by centrifugation at 15,000 x *g* for 10 min at 4 °C. Lysates were pre-cleared with 30 µl of a 50% slurry of protein A (Invitrogen, #15918–014) and protein G agarose beads (Invitrogen, #15920–010) by incubation for 20 min at 4 °C. Pre-cleared lysates were then incubated with 0.7 µg of rabbit anti-GFP antibody for 2 hours at 4 °C to precipitate Robo1-GFP. After incubation, 30 µl of a 50% slurry of protein A and protein G agarose beads was added and samples were incubated for an additional 30 min at 4 °C. The immunocomplexes were washed three times with lysis buffer, boiled for 10 min in 2 x Laemmli SDS sample buffer (Bio-Rad, #1610737) and analyzed by western blotting. Proteins were resolved by SDS-PAGE and transferred to nitrocellulose membrane (Amersham, #10600032). Membranes were blocked with 5% dry milk and 0.1% Tween 20 in PBS for 1 hr at room temperature and incubated with primary antibodies overnight at 4 °C. Following three washes with PBS/0.1% Tween 20, membranes were incubated with the appropriate HRP-conjugated secondary antibody at room temperature for 1 hr. Signals were detected using Clarity ECL (Bio-Rad, #1705061) and SuperSignal West Femto Maximum Sensitivity Substrate (Thermo Fisher, 34094) according to manufacturer's instructions. Antibodies used: for immunoprecipitation, rabbit anti-GFP and for western blot, rabbit anti-GFP (1:500, Invitrogen, #a11122), mouse anti-Myc (1:1000, DSHB, #9E10-C), mouse anti-HA (1:1000, Biolegend, 901502), mouse anti-FLAG (1:1000, Sigma, F1804-50ug), mouse anti-beta-tubulin (1:1000, DSHB, #E7) and HRP goat anti-mouse (1:10,000, Jackson Immunoresearch, #115-035-146).

To examine the biochemical interaction of Robo1, Comm, and Nedd4 in vivo, lysates were prepared from wild-type embryos as well as those expressing 5 X UAS Comm-myc WT with or without 10 X UAS Nedd4-HA under the *elav gal4* driver. To prepare lysates from *Drosophila* embryos, approximately 100 µl of embryos (from egg-laying plates incubated for 18–24 hr at 25 °C) were treated with 50% bleach (Essendant) for 2 min and washed three times with embryo wash buffer (120 mM NaCl supplemented with 0.1% (v/v) Triton X-100, Sigma). Embryos were then rinsed with ice cold TBSV buffer (150 mM NaCl,10 mM Tris pH 8.0, 2 mM Sodium orthovanadate), and lysed with manual homogenization using plastic pestles in 300 µl lysis buffer (TBSV supplemented with 1% Surfact-AMPS NP40 and 1 x complete protease inhibitors). Homogenized samples were then incubated for 30 min at 4 °C with agitation and recovered by centrifugation at 15,000 rpm for 15 min 4 °C.

Immunoprecipitation for myc (Comm) was then conducted using the Pierce anti c-myc magnetic bead kit (Thermo Fisher, 88844) according to the manufacturer's optimized protocol. Lysates were incubated with beads at RT for 30 min and washed twice with the included wash buffer. Immunocomplexes were then eluted from the beads via 10 min RT incubation with the provided acidic elution buffer. This elution was then neutralized with 15 µM Tris pH 8.5 and boiled for 10 min with 4 X Laemmli Sample Buffer (Bio-Rad 1610747).

Proteins were resolved by SDS-PAGE and transferred to nitrocellulose membrane (Amersham, #10600032). Membranes were blocked with 5% dry milk and 0.1% Tween 20 in PBS for 1 hr at room temperature and incubated with primary antibodies overnight at 4 °C. Following three washes with PBS/0.1% Tween 20, membranes were incubated with the appropriate HRP-conjugated secondary antibody at room temperature for 1 hr. Signals were detected using Clarity ECL (Bio-Rad, #1705061) and SuperSignal West Femto Maximum Sensitivity Substrate (Thermo Fisher, 34094) according to manufacturer's instructions. Antibodies used for western blot: Mouse anti-Robo (1:50, DHSB, 13C9), rabbit anti-Myc (1:1000, Sigma C3956-2MG), mouse anti-HA (1:1000, Biolegend, 901502), mouse anti-beta-tubulin (1:1000, DSHB, #E7) and HRP goat anti-mouse (1:10,000, Jackson Immunoresearch, #115-035-146) and goat anti-rabbit (1:10.000 Jackson Immunoresearch, 111-035-003).

## Immunostaining

Transiently transfected COS-7 cells were washed once with ice-cold PBS, fixed for 15 min in 4% paraformaldehyde at room temperature, permeabilized with 0.1% Triton X-100 in PBS (PBT) for 10 min and then blocked in PBT + 5% NGS (normal goat serum) for 30 min at room temperature. Cells were then incubated with primary antibodies diluted in PBT + 5% NGS overnight at 4 °C. After three washes in PBT, secondary antibodies diluted in PBT + 5% NGS were added and incubated for 1 hr at room temperature. After secondary antibodies, cells were washed three times in PBS and coverslips were mounted in Aquamount. The following antibodies were used: Rabbit anti-Myc (1:500, Sigma,

C3956-2MG), mouse anti-HA (1:500, BioLegend # 901502), Alexa488 Goat anti-mouse (1:500, Invitrogen, #A11029), and Cy3 goat anti-rabbit (1:500, Jackson Immunoresearch, 111-165-003).

Dechorionated, formaldehyde-fixed *Drosophila* embryos were fluorescently stained using standard methods. The following antibodies were used: rabbit anti-GFP (1:250, Invitrogen, #a11122), mouse anti-HA (1:500, BioLegend,#901502), rabbit anti-myc (1:500, Sigma, C3956-2MG), chick anti-beta gal (1:500, Abcam, #ab9361),, mouse anti-Robo (1:50, DSHB, #13C9), mouse anti BP102 (1:50, DHSB, BP102), Alexa647 goat anti-HRP (1:500, Jackson Immunoresearch, #123-605-021), Alexa488 goat anti-rabbit (1:500, Invitrogen, #A11034), Alexa488 goat anti-mouse (1:500, Invitrogen, #A11029), Alexa488 goat anti-chick (1:500, Invitrogen, #A11039), Cy3 goat anti-mouse (1:500, Jackson Immunoresearch, #115-165-003), Cy3 goat anti-Chick (1:500, Abcam, #ab97145), Cy3 goat anti-rabbit (1:500, Jackson Immunoresearch, 111-165-003) and 647 goat anti-HRP (1:1,000, Jackson Immunoresearch, #123-605-021). Embryos were filleted and mounted in 70% glycerol/1xPBS. Surface staining of endogenous Robo1 in *Drosophila* embryos was carried out as previously described (*Bashaw, 2010*). Briefly, embryos were dissected live, blocked with in 5% normal goat serum (NGS) in PBS for 15 min at 4 °C and stained with mouse anti-Robo (1:50, DSHB, #13C9) in PBS for 30 min at 4 °C. Following washes with PBS, embryos were fixed in 4% paraformaldehyde (Electron Microscopy Services, #15710) for 15 min at 4 °C. Following washes with PBS, fixed embryos were then permeabilized with 0.1% Triton X-100 in PBS (PBT) for 10 min and stained with rabbit anti-myc (1:500, Sigma, C3956-2MG) and 647 goat anti-HRP (1:1,000, Jackson Immunoresearch, #123-605-021) in 5% NGS in PBT for one hour at room temperature. Following washes with PBT, secondary antibody consisting of Alexa488 goat anti-mouse (1:500, Invitrogen, #A11029) and Cy3 goat anti-rabbit (1:500, Jackson Immunoresearch, 111-165-003) diluted in 5% NGS in PBT was added and incubated for 1 hr at room temperature. Embryos were then washed with PBT and mounted in 70% glycerol in PBS.

Fixed coverslips of COS-7 cells and samples of *Drosophila* embryo nerve cords were imaged using a spinning disk confocal system (Perkin Elmer) built on a Nikon Ti-U inverted microscope using a Nikon 40 X or60X objective with Volocity imaging software. Images were processed using NIH ImageJ software.

## Quantification and statistical analysis

For analysis of *Drosophila* nerve cord phenotypes, image analysis was conducted blind to the genotype. Data are presented as mean values ± 95% confidence interval. For statistical analysis, comparisons were made between two groups using the Student's *t*-test. For multiple comparisons, significance was assessed using one-way ANOVA with Tukey's *post hoc* tests. Differences were considered significant when $p < 0.05$.

For quantification of biochemical data, blots were imaged with (what's the name of the instrument) and quantification was performed using the densitometry function on ImageJ. Comm, Robo1, and Nedd4 levels in S2$R$ + cells or fly lysates are normalized to tubulin. For immunoprecipitation experiments, ubiquitinated Robo1 was normalized to IGG, lysate Robo1, and tubulin. Nedd4 complex incorporation was normalized to immunoprecipitated Robo1 and lysate Nedd4. Comparisons between two groups were made using Student's T test, and comparisons between multiple groups were made using one-way ANOVA with Tukey's post-hoc test. Differences were considered significant when $p < 0.05$.

For quantification of intracellular localization in COS-7 cells, images of seven cell fields per condition were taken at random locations on the coverslip. Cells with overly bright staining, in which we were unable to un-ambiguously observe intracellular localization, were excluded from analysis. Only cells double positive for both HA and Myc were scored. Cells were scored blind to condition. Distribution of exclusively punctate vs diffuse localization was compared between groups using Chi Square.

For quantitation of Robo1 and Comm Levels in the nerve cord in *Drosophila* embryos, images were first batch processed using a macro in image J that set the channels of all images to common thresholds. These processed images were then analyzed in CellProfiler. A common thresholding algorithm was used on smoothened images of the HRP channel to create masks for the nerve cord and axons. To quantify Robo1, total Robo1/488 intensity was measured within the axon mask and divided by area of the mask. To quantify Comm, total Myc/Cy3 intensity was calculated within the nerve cord mask and divided by area of the mask. Data are presented as mean values ± 95% confidence interval. For statistical analysis, comparisons were made between two groups using the Student's *t*-test, and between

multiple groups using ANOVA with Tukey's post-hoc test. Differences were considered significant when p<0.05.

For quantification of cell body and axonal Comm localization, images were first batch processed using a macro in image J that set the channels of all images to common thresholds. These processed images were then analyzed in CellProfiler. A common smoothing algorithm was applied to all images in the 488 channel (showing GFP-labelled apterous neurons). On this smoothened image, both axons and cell bodies were automatically identified using an algorithm that detects objects based on thresholding and size. This set of objects was then manually edited to remove objects detected outside the appropriate area (posterior 8 segments of the nerve cord). Using the original un-smoothened image in the 488 channel as a guide, objects representing cell bodies were selected manually to generate a 'cell body' mask. An 'axon' mask was generated by subtracting the manually-selected cell bodies from the total object set. To quantify Comm expression in either cell bodies or axons, Cy3 fluorescence was measured within the appropriate mask and divided by the area of those mask. The ratio of axonal Comm to cell body Comm was then calculated. Comparisons between groups were made using one-way ANOVA with Tukey's Post-hoc test and data are represented as mean values +/-95% confidence intervals. Differences were considered significant when p<0.05.

For quantification of colocalization of Comm variants and Robo1 in apterous neurons, images were first batch processed using a macro in image J that set the channels of all images to common thresholds. These processed images were then analyzed in cell profiler. To generate a mask of the area of Robo1 expression, a common smoothing algorithm was applied to all images in the 488 channel (for Robo1). Areas of Robo1 expression, which included both axons and cell bodies, were identified automatically via an algorithm that recognizes objects based on brightness threshold, size, and shape. Using the original un-smoothened image in the 488/Robo1 channel as a guide, this set of automatically identified objects was then edited manually (to exclude artifacts that were not cell bodies or axons) and cell bodies were selected manually. The final edited set of objects was then used to generate masks of the total area of Robo1 expression (consisting of both axons and cell bodies), or that in cell bodies alone. To enhance both Robo1 and Comm puncta, an algorithm was applied in both the 488 (Robo1) and Cy3 (Comm) channels. Using the area of these enhanced images (within either the cell body mask, or total expression area mask) Robo1 colocalization with Comm was measured using Manders Coefficient. Comparisons between groups were made using one way ANOVA with Tukey's post-hoc test. Data are presented as mean values +/-95% confidence intervals. Differences were considered significant when p<0.05.

For quantification of colocalization of Comm variants and late endosomal markers in apterous neurons, images were first batch processed using a macro in image J that set the channels of all images to common thresholds. These processed images were then analyzed in cell profiler. To generate masks of cell bodies, a common smoothing algorithm was applied to all images in the 488 channel (for either Rab7 GFP or Lamp1 GFP), and cell bodies were identified automatically via an algorithm that recognizes objects based on brightness threshold, size, and round shape. To generate a mask for the total area of Comm expression, a common smoothing algorithm was applied to all images in the Cy3 Channel (for Comm). Areas of Comm expression within the nerve cord (cell bodies as well as axons if Comm had axonal expression) were identified automatically via an algorithm that recognizes objects based on size and brightness threshold. To eliminate objects detected outside the appropriate area (for example, areas of Comm expression in motor neurons as opposed to the nerve cord, or in the anterior four body segments) objects were then manually edited using the original un-smoothened image in the Cy3 channel. The final edited set of objects was then used to generate a mask. To enhance both endosomal marker and Comm puncta, an algorithm was applied in both the 488 and Cy3 channels. Using the area of these enhanced images (within either the cell body mask, or total comm expression mask), Comm colocalization with late endosomal markers was measured using Pearson's Coefficient. Comparisons between groups were made using one way ANOVA with Tukey's post-hoc test. Data are presented as mean values +/-95% confidence intervals. Differences were considered significant when $P<0.05$.

To quantify nerve cord crossing phenotypes in embryos expressing Comm and Nedd4, images were first processed by applying automatic Brightness/Contrast optimization to micrographs of nerve cords stained with HRP in ImageJ. To determine nerve cord width, images were then blinded and the widths of the eight posterior-most body segments (measured at their widest point) were measured

manually, and the average of these widths was calculated. These widths were then binned into the following phenotypic categories:≤25 µM, 25<x < 30 µM, and ≥30 µM. Distribution of phenotypes between different genotypes was compared using Fisher's exact test with Freeman-Halton extension, using the raw count of embryos within each phenotypic class. Differences were considered significant if p<0.05.

To measure nerve cord segment collapse, an analysis pipeline was executed in CellProfiler. To generate a mask of the axonal scaffold, a smoothing filter was first applied to the pre-processed images of nerve cords stained with HRP. Then, areas of axonal scaffold were first detected automatically using an algorithm that identifies objects based on brightness, size, and shape. This total set of objects was then converted into a binary thresholded image of the axonal scaffold. Negative spaces in the nerve cord were detected from an inverted version of this thresholded image, also using an automatic object identification algorithm. This set of objects was then manually edited to remove artifacts, and to specifically select negative spaces in the center of body segments, using the original HRP nerve cord image as a guide. The percent of collapsed segments was calculated using the following formula: (# segments lacking negative space)/(total segments)*100. For these calculations, only 'complete' body segments containing both 'rungs' in the thresholded image of the axonal scaffold were used. Embryos were binned into the following phenotypic categories: no collapse (0% collapse), partial collapse (0<x < 100% collapse), and complete collapse (100% segments collapsed). Distribution of phenotypes between different genotypes was compared using Fisher's exact test with Freeman-Halton extension, using the raw count of embryos within each phenotypic class. Differences were considered significant if p<0.05.

## Acknowledgements

We acknowledge past and present members of the Bashaw lab for thoughtful discussions and comments on this manuscript. We especially thank Dr. Madhavi Gorla for guidance on cell and biochemical assays, Dr. Yixin Zang for guidance on the embryo live-dissection experiments and Dr. Camila Barrios Camacho for help with Cell Profiler analysis. The *smurf 15* C fly line were a kind gift from Dr. Haiwei Pi. This research was supported by NIH grants 5T32GM007229-42 and F31NS115345-01 to KS, and NSF Grant IOS-1853719 and NIH Grants R35 NS097340 and R01-HD-105946 to G.J.B

## Additional information

### Funding

| Funder | Grant reference number | Author |
|---|---|---|
| National Institutes of Health | R35-NS-097340 | Greg J Bashaw |
| National Institutes of Health | R01-HD-105946 | Greg J Bashaw |
| National Institutes of Health | F31NS115345 | Kelly G Sullivan |
| National Institutes of Health | 5T32GM007229-42 | Kelly G Sullivan |
| National Science Foundation | IOS-1853719 | Greg J Bashaw |

The funders had no role in study design, data collection and interpretation, or the decision to submit the work for publication.

### Author contributions

Kelly G Sullivan, Conceptualization, Data curation, Formal analysis, Investigation, Methodology, Writing - original draft, Writing – review and editing; Greg J Bashaw, Conceptualization, Resources, Supervision, Funding acquisition, Investigation, Project administration, Writing – review and editing

## Author ORCIDs

Greg J Bashaw ⓘ https://orcid.org/0000-0002-6146-0962

Reviewer #1 (Public review): https://doi.org/10.7554/eLife.92757.3.sa1
Reviewer #2 (Public review): https://doi.org/10.7554/eLife.92757.3.sa2
Author response https://doi.org/10.7554/eLife.92757.3.sa3

## Additional files

### Supplementary files

MDAR checklist

### Data availability

All data generated or analyzed during this study are included in the manuscript and supporting files. Confocal stacks were collected using a spinning disk confocal system (Perkin Elmer) built on a Nikon Ti-U inverted microscope with Volocity imaging software. Images were processed using NIH ImageJ and Adobe Photoshop software. Image analysis was conducted using CellProfiler 4.2.5. All statistics and graphs were generated using GraphPad Prism 9.

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
