## [Editor Report · eLife Assessment]

This work is of **fundamental** significance to the field of nervous system development as it advances our mechanistic understanding of axon guidance. The rigorous biochemical and genetic approaches are **compelling**, experiments are well-controlled, and the major claims are supported by **convincing** data. The study should be of general interest to the developmental neurobiology community.

---

## [Referee Report · Reviewer #1 (Public review)]

Summary:

This study is focused an important aspect of axon guidance at the central nervous system (CNS) midline: how neurons extend axons that either do or do not cross the CNS midline. The authors here address contradictory work in the field relating to how cell surface expression of the slit receptor Robo1 is regulated so as to generate crossed and non-crossed axon trajectories during *Drosophila* neural development. They use fly genetics, cell lines, and biochemical assessments to define a complex consisting of the commissureless, Nedd4 and Robo1 proteins necessary for regulating Robo1 protein expression. This work resolves certain remaining questions in the field regarding midline axon guidance, with strengths out weighing weaknesses; however, addressing some of these weaknesses would strengthen this study.

Strengths:

Strengths include:

- The use of well controlled genetic gain-of-function (over expression) approaches in vivo in *Drosophila* to show that phosphorylation sites (there are 2, and this study allows for assessment of the contributions made by each) in the commissureless (Comm) protein are indeed required for Comm function with respect to regulating axon midline guidance via their role in directing Comm-mediated Robo1 ubiquitination and degradation in the lysosome.

- The demonstration that in vitro, and in a sensitized genetic background in vivo, the Nedd4 ubiquitin ligase regulates Robo1 protein cell surface distribution and also midline axon crossing in vivo.

- Important evidence here that serves to resolve many questions raised by previous studies (not from these authors) regarding how Robo1 is regulated by Comm and Nedd4 family ubiquitin ligases. Further, these results are likely to have implications for thinking about the regulation of midline guidance in more complex nervous systems.

Weaknesses:

- A weakness beyond the purview of revision but important to mention is that the authors chose not to complement their GOF experiments with gene editing approaches to generate endogenous PY mutant alleles of Comm that might have been useful in genetic interaction experiments directed toward revealing roles for endogenous Comm in the regulation of Robo1.

Comments on revised version:

In this revised manuscript the authors provide new experiments and also reasonable explanations to address concerns raised in the initial review. I am satisfied that these efforts address satisfactorily the points raised in the initial review and that this study has been strengthened. This is an interesting body of work that adds to our understanding of CNS midline guidance molecular mechanisms.

---

## [Referee Report · Reviewer #2 (Public review)]

Summary:

Sullivan and Bashaw delve into the mechanisms that drive neural circuit assembly, and specifically, into the regulation of cell surface proteins that mediate axon pathfinding. During nervous system development, axons must traverse a molecularly and physically complex extracellular milieu to reach their synaptic targets. A fundamental, conserved repulsive signaling pathway is initiated by the Slit-Robo ligand-receptor pair. Robo, expressed on axon growth cones, binds Slit, secreted by midline cells, to prevent "pre-crossing" and "re-crossing" of axons at the midline. To control this repulsion, Robo surface levels are tightly regulated. In *Drosophila*, Commissureless (Comm) downregulates Robo surface levels and is required for axon crossing at the midline. Several studies suggest that PY motifs in Comm are required to localize Robo to endosomes. PY motifs have been shown to bind WW-domain containing proteins including the ubiquitin ligase Nedd4 family, so the authors propose that Comm may regulate Robo through Nedd4 interactions. Previous studies have hinted at a role for Nedd4-mediated ubiquitination of Comm in regulation of Robo localization, but there have also been conflicting data. For example, Comm mutants that are unable to be ubiquitinated mimic wild-type Comm, suggesting that ubiquitination of Comm is not required for regulation of Robo. The current study utilizes a suite of genetic analyses in *Drosophila* to resolve discrepancies pertaining to the mode of Comm-dependent regulation of Robo1 and proposes that Comm acts as an adapter for the Nedd4 ubiquitin ligase to recognize Robo1 as a substrate. The authors also demonstrate that Nedd4 is indeed required for midline crossing.

Strengths:

While this work is more incremental rather than field-shifting, it is nonetheless an excellent example of a rigorous, thorough analysis that culminates in enriching our mechanistic understanding of how neurons regulate cell-surface receptors in a spatiotemporal manner to control fundamental steps of circuit wiring. The experimental approach is thorough, and the manuscript is extremely well-written.

Weaknesses:

Some key experiments (eg. complex formation) were performed in cell culture in an overexpression background. However, updated experiments demonstrated complex formation using immunoprecipitation in tissues overexpression the corresponding components. Also, there was a missed opportunity to bolster the model proposed by using Comm PY mutants in several experiments.

Comments on revised version:

The revised manuscript bolsters the authors' conclusions and now provides evidence for interactions in tissue. No additional experiments are needed.

---

## [Author Response]

The following is the authors’ response to the original reviews.

**Response to Editor and Reviewer Comments:**

Many thanks to the editor and reviewers for the thoughtful assessment of our manuscript “Commissureless acts as a substrate adapter in a conserved Nedd4 E3 ubiquitin ligase pathway to promote axon growth across the midline.” Thank you also for the positive comments about the quality of our writing, and for deeming our study rigorous and thorough. We are very pleased that, overall, you believe our combination of genetic and biochemical approaches offers useful insight into the mechanism of Robo regulation at the *Drosophila* embryonic midline and effectively reconciles the contradictory findings of previous studies done in this field.

**Response to the previous Public Reviews:**

We appreciate the concerns expressed by the reviewers and the suggestions of areas in which the study and manuscript could be improved. The reviewer suggestions were very helpful as we revised our manuscript in order to strengthen our mechanistic understanding of Robo downregulation and better characterize the role Nedd4 plays in this process. We strongly agree with Reviewer 1 that our insight into the mechanism of Robo downregulation via Comm would be much stronger had we not solely relied on overexpression experiments to investigate the effects of PY motif mutations on Comm function. While it is outside the scope of this particular paper, we appreciate your suggestion to use gene editing to investigate the role of PY motif mutation on endogenous comm function and believe this would be a useful question to address in future papers. In addition to this concern, both reviewers identified additional opportunities to strengthen the paper. We have done our best to incorporate reviewer suggestions and will outline how we addressed the following four areas that were identified by both reviewers as areas where additional data could strengthen our conclusions:

(1) Additional experiments to examine Comm and Robo1 localization in vivo*:* Characterizing Robo localization in vivo when co-expressed with PY-mutant Comm variants.

(2) Testing biochemical interactions in embryonic protein extracts: Examining the biochemical interaction between Robo, Comm, and Nedd4 in a more biologically relevant context than cell culture.

(3) Additional genetic interaction experiments: (A) Investigating whether Nedd4 overexpression enhances the Comm G.O.F phenotype of enhanced ectopic crossing. (B) Testing for additional genetic interactions with *comm*.

(4) Editing the text of the manuscript for clarity.

(1) Characterizing Robo localization in vivo when co-expressed with Comm variants.

In the first version of our manuscript, we characterized the localization of wild-type and PY mutant Comm variants expressed in apterous neurons (Figure 5C), but did not examine how these variants of Comm affected localization of their cargo Robo1. To address this gap, we co-expressed 10X UAS Comm-myc (WT, 1PY, 2PY) with 10X UAS Robo-HA under the ap gal4 driver, visualized Comm and Robo by immunostaining for Myc and HA, and measured colocalization between Comm and Robo. We found that Robo colocalizes equally with all comm variants and that its expression pattern mimics that of the Comm variant with which it is expressed. We observe that Robo is restricted to cell bodies when overexpressed with WT Comm but “leaks out” into axons when co-expressed with Comm 1PY or 2PY. This finding suggests that PY motifs are not only required for effective Comm localization to the appropriate cellular areas, but also for proper routing of its cargo, Robo1. These new data are presented in a new supplemental figure: Figure S3.

(2) Examining the biochemical interaction between Robo, Comm, and Nedd4 in vivo*.*

To examine biochemical interaction between Comm, Robo, and Nedd4 in a more biologically relevant context, we performed immunoprecipitations in fly embryonic lysate prepared from the following categories: WT, elav gal4: 5X UAS Comm-myc WT, and elav gal4: 5X UAS Comm-myc WT + 10X UAS Nedd4-HA. We performed immunoprecipitation for myc (Comm), and blotted for endogenous Robo, Myc (Comm), and HA (Nedd4). Corroborating our results in cell culture (Figure 7 A-C), we were able to pull down a three-protein complex consisting of Comm, Nedd4 and Robo in embryonic fly tissue. These new data are presented in a new supplemental figure: Figure S8.

(3) Investigating additional genetic interactions between Comm and Nedd4.

(A) In our submitted manuscript, we demonstrated that overexpression of Nedd4 enhances Comm-induced downregulation of Robo levels (Figure 7 D-G). To determine whether Nedd4 also increases ectopic crossing, which is a morphological output of Comm activity/Robo downregulation, we analyzed nerve cord phenotypes in embryos from the following categories: WT, embryos expressing WT Comm under the elav gal4, and embryos co-expressing WT Comm and Nedd4 under the elav gal4 driver. We measured nerve cord widths and sorted them into three different “bins” of phenotypic severity, with more severe phenotypes being characterized by thinner nerve cords. We find that the distribution of phenotypes in embryos expressing Comm alone differs significantly from embryos expressing Comm + Nedd4, with the latter shifted toward more severe/thinner phenotypic classes. In addition to examining nerve cord width, we investigated whether Nedd4 can enhance collapse of the nerve cord segments (defined by loss of negative space within the segment) induced by Comm overexpression. We determined percentage of collapsed nerve cord segments and divided these values into three phenotypic classes: no collapse, partial collapse, and total collapse. The distribution of phenotypes in embryos co-expressing Nedd4 and Comm differs significantly from those expressing Comm alone. In the Comm expressing population, we only observe nerve cords with no or partial collapse, but in flies co-expressing Comm and Nedd4 we observe the more severe complete collapse phenotype. These findings suggest that addition of Nedd4 enhances the Comm gain of function phenotype both by further reducing nerve cord width and increasing the occurrence of defects related to ectopic crossing. These new data are presented in a new supplemental figure: Figure S9.

(B) The reviewers also suggested additional genetic interaction experiments between Nedd4 and Comm. It was suggested that we included experiments to look at Nedd4 manipulations in Comm null mutant backgrounds. However, given the complete penetrance and expressivity of the Comm null mutation in which no axons cross the midline, these experiments would not be informative. As an alternative, we attempted to use the described hypomorphic Comm allele, but here too, the baseline commissural axon guidance defects are too strong to allow meaningful detection of enhanced phenotypes. Finally, we tested whether removing one copy of *comm* could reveal phenotypes in the *nedd4* zygotic mutants, but we did not detect defects. This is perhaps unsurprising given that comm heterozygotes have no detectable midline crossing defects.

(4) Text edits.

We have made a variety of changes to decrease ambiguity in the text and create a more user-friendly experience for the reader. In the text, as opposed to just the figures, we now explicitly state whether we use 5X or 10X UAS constructs for each of our overexpression constructs. We also edited all mentions of the truncated frazzled construct (FraDc) so that they are uniform. We have also edited all mentions of MiMIC so that they are uniform. In addition, we answer a few questions the reviewers posed. First, we clarify that S2R+ cells express endogenous Comm at very low levels. In addition, we clarify about how we know expression levels are similar across the three Comm variants by explaining that transgenes incorporated into the fly genome by targeted insertion into the same location on the third chromosome.

We hope that these changes adequately address reviewer concerns, strengthen our study, and enhance readability of the paper. We appreciate the time you took to evaluate our manuscript and the thoughtful commentary and suggestions that you provided.